# Bond percolation in coloured and multiplex networks

Ivan Kryven [1]

Percolation in complex networks is a process that mimics network degradation and a tool that reveals peculiarities of the network structure. During the course of percolation, the emergent properties of networks undergo non-trivial transformations, which include a phase transition in the connectivity, and in some special cases, multiple phase transitions. Such global transformations are caused by only subtle changes in the degree distribution, which locally describe the network. Here we establish a generic analytic theory that describes how structure and sizes of all connected components in the network are affected by simple and colour-dependent bond percolations. This theory predicts locations of the phase transitions, existence of wide critical regimes that do not vanish in the thermodynamic limit, and a phenomenon of colour switching in small components. These results may be used to design percolation-like processes, optimise network response to percolation, and detect subtle signals preceding network collapse.

---

[1] Van't Hoff Institute for Molecular Sciences, University of Amsterdam, PO Box 941571090 GD Amsterdam, The Netherlands. Correspondence and requests for materials should be addressed to I.K. (email: ikryven@gmail.com)

One of the richest tools for exploring properties of complex networks is probing these networks by randomly removing links. This process is called percolation, and on many occasions percolation has shaped our understanding of physical and social phenomena[1]. Naturally, percolation is related to network resilience. This connection was exploited, for instance, in studies on communication, transportation and supply networks[2–5]. Percolation has defined the modern view on disease epidemics[2,6–9], as well as on other spreading processes[10–12]. Percolation is instrumental in material science, where the gelation[13–16] and jamming[17,18] have been both connected to this process.

It is known that during percolation, complex networks undergo a series of non-trivial transformations that include splitting of connected components and criticality in the global connectivity[5,19–22]. Even the simplest models that define networks solely by their degree distribution (the configuration models) do also feature these phenomena. In many studies, this observation justified the use of the configuration model as the null-model, which one would compare against any real network, in order to reveal presence of trends that are, or are not, explainable by the degree distribution alone.

In the edge-coloured configuration model, also known as the network of networks[21] or multiplex[23], the edges are categorised into different layers or colours. This distinction leads to a more realistic representation of complex networks. Indeed, most of real networks do feature different types of interactions: chemical bonds, communication channels, social contacts between infected individuals, among other examples, all require a partition of the edges into discrete categories[9,23–26]. Furthermore, a whole world of possibilities unfolds when the number of colours is virtually unlimited: different edge colours may encode all combinations of one interaction occurring between different types of nodes, as was proposed in the assortative mixing model[27], or one type of interaction with discrete strengths[28]. In modular networks, interactions within each community can also be represented with a unique colour. In contrast to uni-coloured networks[29], less is known about how percolation happens in coloured networks. For instance, presence of multiple phase transitions has been reported[20,21,23], but no complete theory could explain their nature nor predict locations where these phase transitions occur. It has been observed that negative correlation in the degree distribution may lead to multiple phase transitions[23].

This paper demonstrates that in order to explain percolation in coloured networks, one has to study not only the giant component but also the rest of the system. In fact, the tail asymptote for the sizes of connected components already covers sufficient amount of information. By building upon this result, we establish universal criteria of phase transitions for two percolation processes: simple bond percolation, which removes every edge with equal probability, and the colour-dependent bond percolation, in which edges of different colours have different probability to be removed. This theory detects multiple phase transitions, if such occur, and in the case of colour-dependent percolation, the theory yields a manifold containing all critical points. Besides critical points, the theory also predicts existence of wide critical intervals that do not vanish in infinite systems and colour switching that occurs in small connected components during percolation.

that bear half edges as defined by the degree distribution. These half edges are then joined randomly, so that a pair of half edges belonging to different nodes becomes an edge. By treating any configuration of such matchings as equiprobable we can then ask various questions about the relationship between the degree distribution (the only parameter of the model) and emergent properties of the network. The question of characterising sizes of connected components often arise in theoretical and applied studies. Newman et al.[19] observed that in configuration networks, the size distribution of connected components can be found numerically, and these ideas have been later formalised into an analytical theory[30] (see the Methods section for a brief summary). Interesting developments of the configuration model can be obtained by constraining the configurations with extra requirements. For example, by fixing the clustering coefficient[31], number of triangles[32] or adding directionality to edges[33]. In edge-coloured configuration model, edges are labelled with an arbitrary number of colours, see Fig. 1b. That is to say, we assign labels 1, 2, …, N to the half edges and then constrain the configurations of edges to join only the matching colours (see Fig. 1b). Network models that accommodate edges of different types, also known as multilayer networks, are at the core of contemporary network science as they feature better predictions[34]. As a demonstration, Supplementary Note 1 discuses several examples of popular network problems that can be non-trivially mapped to the edge-coloured configuration model. In the same time, the edge-coloured configuration model features a richer spectrum of behaviour that is not observed in unicoloured networks. For instance, cascade events my occur under strong notion of connectivity[22,34–36]. There are many studies that focus on important special cases, and apply the multiplex formalism to explain phenomena in empirical data[25,35,37–40]. However, the theoretical aspects are less developed in these studies as they scatter between special cases to explain empirical observations rather than to develop a single universal theory.

The current paper presents a concise analytical theory for sizes of connected components. In this theory $N$ can be an arbitrarily large number, and the key equations are presented in terms of easy-to-use matrix algebra expressions. Surprisingly, even though the input for the edge-coloured configuration model is a multivariate degree distribution, for most of the results it suffices to know only the first mixed moments of this distribution, which practically means that one can easily characterise the size distribution of connected components in networks with thousands of colours.

**Asymptotic theory for edge-coloured configuration model.** More formally, suppose that every edge is assigned one of $N$ colours, so that a randomly chosen node bears $k_1$ edges of colour one, $k_2$ edges of colour two, and so on. A state of a node is parametrised by a vector of colour counts $\boldsymbol{k} = (k_1, …, k_N)$, see Fig. 1c. Let $u(\boldsymbol{k})$ be the probability that a randomly chosen node has state $\boldsymbol{k}$. The expected value of function $f(\boldsymbol{k})$ with respect to probability distribution $u(\boldsymbol{k})$, is defined by the following sum:

$$\mathbb{E}[f(\boldsymbol{k})] := \sum_{\boldsymbol{k} \geq 0} f(\boldsymbol{k}) u(\boldsymbol{k}). \tag{1}$$

## Results
**Connected components in random networks.** In conventional configuration model, see Fig. 1a, one starts with a set of nodes

For most of the results derived in this paper it is enough to know a relatively small portion of information that describes $u(\boldsymbol{k})$, namely, this information is contained in $\boldsymbol{\mu}_0$, $\boldsymbol{M}$

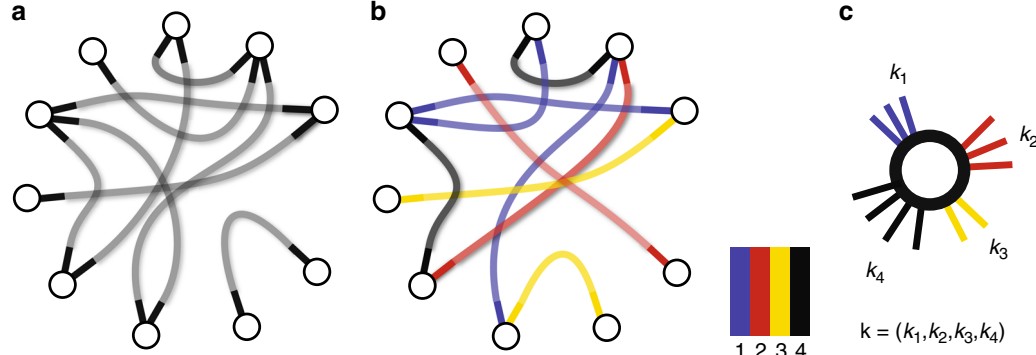

**Fig. 1** The concept of edge-coloured configuration model. **a** The configuration model with unicoloured edges[19]: any configuration of edges that matches all pairs of half-edges produces a valid network. **b** Edge-coloured configuration model: valid configurations have to link matching pairs of colours. In this example, $N = 4$ colours are used as indicated by the colour palette, although the methodology of this paper allows $N$ to be an arbitrary number. **c** The configuration of a randomly chosen node is a vector of colour counts

and $T_i$ as defined via expectations of $u(k)$ in the following way:

$$\mu_0 = (\mathbb{E}[k_1], \mathbb{E}[k_2], \ldots, \mathbb{E}[k_N])^\top, \tag{2}$$

$$M_{i,j} = \frac{\mathbb{E}[k_i k_j]}{\mathbb{E}[k_j]} - \delta_{i,j}, \; i, j = 1, \ldots, N, \tag{3}$$

$$(T_i)_{j,l} = \frac{\mathbb{E}[k_i k_j k_l]}{\mathbb{E}[k_i]} - \frac{\mathbb{E}[k_i k_j]\mathbb{E}[k_i k_l]}{\mathbb{E}[k_i]^2}, \; i, j, l = 1, \ldots, N. \tag{4}$$

In this work, we use weak connectivity notion as the working definition of a connected component. This means that two nodes are considered to be connected if there is a path that joins them and this path may combine any colours. For example, in Fig. 1b we have two connected components. Note, there are several other ways to define a connected component, all of which lead to a different asymptotic theory and percolation properties, for example: strong, in-, and out-components[19], colour-avoiding components[38] and mutually dependent components[21,22,35,36]. In our previous work[33], the formal expression for the size distribution of connected components $w(n)$ is derived by applying Joyal's theory of species,

$$w(n) = \sum_{\substack{k_1 + \ldots + k_N = n - 1 \\ k_i \geq 0}} \left( \widetilde{D} * u * u_1^{*k_1} * \ldots * u_N^{*k_N} \right)(k), \tag{5}$$

where operations $f * g$ and $f^{*k}$ denote, respectively, the $N$-dimensional convolution product and the convolution power[33]. The interpretation of Eqs. (3) and (4) and the definition of the auxiliary funciton $\widetilde{D}$ is given in the Methods section. Although the Eq. (5) is mathematically robust, it cannot be practically computed even when $N$ is moderately large due to severe computational complexity of $O(n^N \log n)$. This issue can be surpassed by deriving (see the Methods section) the tail asymptote of Eq. (5) for arbitrary number of colours $N$ and non-scale free degree distribution:

$$w_\infty(n) = C_1 n^{-3/2} e^{-C_2 n}, \; n \gg 1, \tag{6}$$

where coefficients $C_1 > 0$, $C_2 \geq 0$ are defined in terms of $\mu_0$, $M$ and $T_i$ (the exact expressions are given in the Methods section). Here,

$C_2 \gg 0$ implies exponentially rapid decrease of the asymptote; close-to-zero values of $C_2$ are associated with transient scale-free behaviour that is eventually overrun by an exponentially fast decrease, and $C_2 = 0$, triggers a scale-free behaviour. In fact, as shown in the Methods section, the scale-free form occurs only when the following criterion holds true:

$$\mathbf{v} \in \ker[M - I], \; \text{and} \; \frac{\mathbf{v}}{|\mathbf{v}|} > 0, \tag{7}$$

where $|v| = \sum_i v_i$. This is the necessary and sufficient condition for criticality. Recall that $\ker[A]$ denotes all vectors $\mathbf{v}$ for which $A\mathbf{v} = 0$. Eq. (7) is the $N$-colour generalisation to the famous Molloy and Reed criterion[41]. Indeed by setting $N = 1$ one recovers $\mathbb{E}[k_1^2] - 2\mathbb{E}[k_1] = 0$.

We will now discuss the inner structure of the components. Let a randomly selected component of size $n$ has $v_i$ edges of colour $i$. We found that the vector of colour fractions $f = (f_1, f_2, \ldots, f_N)$, $f_i := \frac{v_i}{n-1}$ is distributed according to the $N$-variate Gaussian distribution:

$$\mathbb{P}[f|n] = \mathcal{N}\left(f, m, \frac{1}{n}\Sigma\right), \; n \gg 1, \tag{8}$$

where the expressions for the mean value vector and the covariance matrix are known functions of $M$ and $T_i$ given in the Methods section.

Despite being so short, Eqs. (6)–(8) together constitute a very rich theoretical result, and most of the remaining text of the paper is devoted to discussing the implications of these equations to network science as well as applications to different types of percolation. Note that all three of these equations are universal: they do not depend on a specific degree distribution and are suitable when working with analytical distributions as well as the empirical ones. So far, similar universality has only been known to hold in unicoloured networks[19,30,42]. A very small portion of information that is encoded in the degree distribution, that is $\mu_0$, $M$ and $T_i$, is important when deriving the asymptotic emergent properties. Moreover, the larger is a connected component, the more it "forgets" about the exact shape of the degree distribution, yet, some information is transmitted to the higher levels of the hierarchy without any loss at all. This information is condensed into parameters $m$, $C_1$ and $C_2$. Furthermore, the mean colour fraction $m$ does not depend on component size: the colour fractions fluctuate around the same mean value in all finite components.

Eqs. (6)–(8) also provide a powerful tool for analysis in the context of evolving networks. In this case, one has to provide the trajectory for an evolving degree distribution. This trajectory may be empirical or driven by a known mechanism. An important example of such a evolution mechanism is a random removal of edges also known as the percolation process. In the course of this process, the parameters that define Eqs. (6)–(8) also evolve, which may lead to interesting dynamics. For instance the network becomes critical every time when $C_2$ vanishes during the evolution of the process. Figure 2 gives a schematic representation for several such scenarios and the next section introduces analytical equations for evolving parameters of Eqs. (6)–(8) as driven by two types of percolation processes. When the degree distribution is altered as a consequence of edge removal, colour fractions $m$ may also change. In the Discussion section we provide several examples where $m$ features an unexpected switching behaviour.

Finally, it needs to be mentioned that the giant component itself is an exception from the general trend featured by finite components: it has a different distribution of colour fractions, and unlike in the case of finite components, the whole degree distribution becomes important.

**Simple bond percolation.** As shown in Eq. (7), there exists a class of degree distributions for which $C_2 = 0$ and therefore the asymptote (6) is scale-free. We call these degree distributions critical. Let us investigate a simple process that continuously changes the degree distribution in such a way that the latter traverses though this critical class. See Fig. 2 for a few conceptual illustrations of how such evolution may look like.

We first consider a process that thinners the network by randomly removing edges with probability $1 - p$, or equivalently, by keeping edges with probability $p$, which is independent of the edge colour. As shown in the Methods section, such removal of edges affects the degree distribution so that it becomes characterised by a new triple $\boldsymbol{\mu}_0', \boldsymbol{M}', \boldsymbol{T}_i'$ that depends on $p$:

$$\boldsymbol{\mu}_0' = p\boldsymbol{\mu}_0, \tag{9}$$

$$\boldsymbol{M}' = p\boldsymbol{M}, \tag{10}$$

$$\left(\boldsymbol{T}_i'\right)_{j,l} = p^2(\boldsymbol{T}_i)_{j,l} + p(1-p)M_{j,i}\delta_{j,l}. \tag{11}$$

By plugging $\boldsymbol{M}'$ into the general criticality criterion (7) one obtains the following $p$-dependent criterion: the edge-coloured network features the critical behaviour at $p = p_c \in (0, 1]$ if there is vector $\mathbf{v}$ for which,

$$\mathbf{v} \in \ker[p_c\boldsymbol{M} - \boldsymbol{I}], \text{ and } \frac{\mathbf{v}}{|\mathbf{v}|} \ge 0. \tag{12}$$

An alternative way of looking at Eq. (12) is reformulating this criterion as an eigenvalue problem. Eq. (12) is satisfied at $p_c = \lambda^{-1}$ if and only if all of the following conditions hold true:

1. $(\lambda, \mathbf{v})$ is an eigenpair of $\boldsymbol{M}$,
2. $\lambda > 1$,
3. $\mathbf{v}$ is non-negative when normalised, i.e., $\frac{\mathbf{v}}{|\mathbf{v}|} \ge 0$.

Note, that if $\boldsymbol{M}$ is not primitive, Eq. (12) may have multiple solutions $p_c$.

**Colour-dependent percolation and critical manifolds.** We will now consider the case when the probability $\mathbf{p} = (p_1, p_2, \ldots, p_N)^\top$ is a vector instead of a scalar. In this more

general setting, $p_i$ are the probabilities that $i$-coloured edge is not removed, so that we have a colour-dependent percolation. Such situation has a clear interpretation: edges of different type may have different susceptibility to damage, or resistance to infection transition. In the Methods section we show that the colour-dependent percolation with parameter $\mathbf{p}$ affects the degree distribution in the following fashion:

$$\boldsymbol{\mu}_0' = \text{diag}\{\mathbf{p}\}\boldsymbol{\mu}_0, \tag{13}$$

$$\boldsymbol{M}' = \text{diag}\{\mathbf{p}\}\boldsymbol{M}, \tag{14}$$

$$\boldsymbol{T}_i' = \text{diag}\{\mathbf{p}\}\boldsymbol{T}_i\text{diag}\{\mathbf{p}\} + \text{diag}\{\mathbf{p}\}\text{diag}\{1 - \mathbf{p}\}\text{diag}\{M_{1,j}, \ldots, M_{N,j}\}. \tag{15}$$

By plugging $\boldsymbol{M}'$ into Eq. (7) one obtains the criterion for colour-dependent percolation: edge-coloured network features the critical behaviour at $0 < \mathbf{p} < 1$ if and only if

$$\mathbf{v} \in \ker[\text{diag}\{\mathbf{p}\}\boldsymbol{M} - \boldsymbol{I}], \text{ and } \frac{\mathbf{v}}{|\mathbf{v}|} \ge 0. \tag{16}$$

The latter criterion can be viewed as a parameter equation for a critical manifold placed in an $N$-dimensional space, and, unlike in the case of simple percolation, one cannot reduce criterion (16) to an eigenvalue problem. In the Discussion section we present a few examples of critical manifolds that have been computed numerically using criterion (16). The other asymptotic properties derived in this work, as for instance the size distribution asymptote (6) and the distribution of colour fractions (8), can be readily computed by plugging expressions (13)–(15) in these equations.

It is important to mention that criticality criteria (7), (12) and (16) contribute to the series of findings that exist in the literature. Note that these criteria provide necessary *and* sufficient conditions for criticality. If one is interested only in the *necessary* condition then it is sufficient to check if the corresponding determinant vanishes. Namely, the necessary condition for criticality (as follows from Eq. (7)) is then given by:

$$\det(\boldsymbol{M} - \boldsymbol{I}) = 0. \tag{17}$$

This equation has also been derived in ref. [43] (Eq. (11)). The other link with previous works emerges if we assume that $\boldsymbol{M}$ is primitive, then (12) reduces to:

$$p_c = \rho(\boldsymbol{M})^{-1}, \tag{18}$$

where $\rho(\cdot)$ denotes the spectral radius. This equation has been also derived in ref. [44] (Eq. (27)) and independently in ref. [23] (Eq. (7)). When $\boldsymbol{M}$ is not primitive, then there is an elementary matrix that transforms $\boldsymbol{M}$ to the block diagonal form, in which case, one may study the blocks separately with Eq. (18). In ref. [36] the authors also use a similar concept to the critical manifolds introduced in Eq. (16) to study cascades of failures on interdependent networks with two colours ($N = 2$). In Supplementary Note 1, several other extensions of the configuration model are reformulated in such a way that the current theory can be applied to these generalisations as well. Most importantly, these include: directed edge-coloured networks (see also Supplementary Figure 1), multi-graphs, multiplex networks with edge overlap, and node-labelled networks[24,45,46].

**Hierarchy of critical points and secondary phase transitions.** By excluding some of the colours, one may define stronger notions of connectivities. Any non-empty subset $S \subset \{1,\ldots,N\}$ gives rise to a

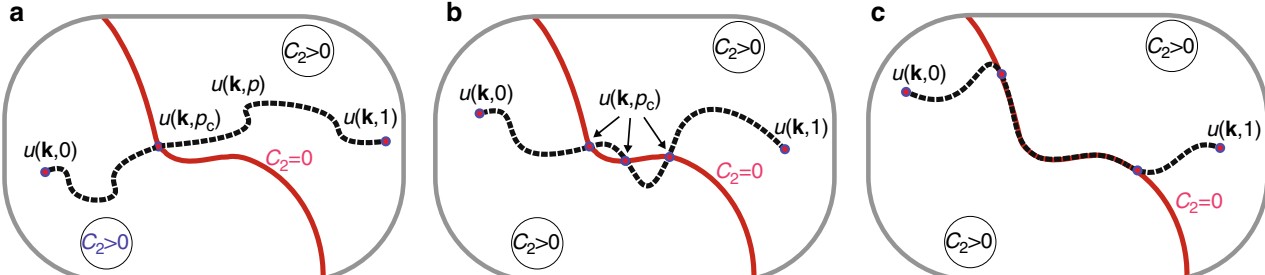

**Fig. 2** Evolution trajectories of the degree distribution during edge removal. An example of a scenario when the network: **a** becomes critical once along the trajectory, **b** becomes critical at three disjoint points, and **c** becomes critical on a continuous interval of the trajectory

valid definition of a path, and all possibilities collected together comprise a power set $\mathcal{P}\{1, \dots, N\}\setminus\{\emptyset\}$, *i.e.* the set of all subsets. Similar considerations were explored in colour-avoiding percolation[38]. This power set features a natural hierarchy as defined by the inclusion relation "$\subset$", and if $S_1 \subset S_2$ then the criticality associated to this colour subsets are strictly ordered $p_{S_2} < p_{S_1}$. The criterion (12) can be readily adjusted to detect criticalities under $S$-connectivity notion. It is enough to replace $M$ by

$$\text{diag}\{\mathbf{x}_S\} M \,\text{diag}\{\mathbf{x}_S\}, \tag{19}$$

where $\mathbf{x}_S$ is the indicator vector for colour subset $S$. In this way, one has means of identifying all critical points $p_S$ and all colour subsets that correspond to them. Although, recovering all $p_S$ requires solving $2^N - 1$ eigenvalue problems as defined by Eq. (19), much can be said about how $p_S$ are distributed when $M$ is a diagonal-dominant matrix. In this case, the critical points are grouped into batches, and the locations of this batches are given by the Gershgorin circle theorem: each critical point $p_S$ belongs to at least one of the following intervals:

$$\left[\max(M_{i,i} + r_i, 1)^{-1}, \max(M_{i,i} - r_i, 1)^{-1}\right], \, r_i = \sum_{j \neq i} M_{i,j}, \, i = 1, \dots, N. \tag{20}$$

The latter relation shows that there is a "downward causation effect": the mutual criticality of many colours sets the boundary on when the criticality of any subset of this colours can occur. Since diagonal dominance of matrix $M$ corresponds to a highly modular structure of the network, we may conclude that the secondary critical points in such networks are not uniformly distributed but appear in groups.

**The giant component and the average component size**. The node-size of the giant component $g_{\text{node}}$ is the probability that a randomly sampled node belongs to the giant component. This quantity is often simply called the size of the giant component and used as a measure of connectedness in sparse networks. As shown in the Methods section, the node-size of giant component can be written in terms of expectations $\mathbb{E}[\cdot]$ of the degree distribution:

$$g_{\text{node}} = 1 - \mathbb{E}[\mathbf{s}^{\mathbf{k}}], \, \mathbf{s} = (s_1, s_2, \dots, s_N)^\top, \tag{21}$$

where elements of vector $\mathbf{s}$ are defined by an implicit relation,

$$s_i = \frac{\mathbb{E}[k_i \mathbf{s}^{\mathbf{k}-\mathbf{e}_i}]}{\mathbb{E}[k_i]}, \, i = 1, \dots, N, \tag{22}$$

and $\mathbf{e}_i$ are the standard basis vectors. In a similar fashion, the edge-size of the giant component (the probability that a randomly

sampled edge of colour $i$ is a part of the giant component) is given by $g_i = 1 - s_i^2$, $\mathbf{g} = (g_1, \dots, g_N)$. By weighting this vector with the total fractions of coloured edges $c_i = \mathbb{E}[k_i]/\sum_{j=1}^N \mathbb{E}[k_j]$, one obtains the vector of colour fractions in the giant component,

$$\mathbf{v}^* = (v_1^*, v_2^*, \dots, v_N^*), \, v_i^* = \frac{g_i c_i}{\mathbf{g}^\top \mathbf{c}}, \tag{23}$$

which is the giant-component analog of vector $\mathbf{m}$ introduced in Eq. (8). Since the giant component is infinite by definition, $\mathbf{v}^*$ is deterministic and does not feature fluctuations.

The weight-average size of finite connected components $w_{\text{avg}} = \frac{\mathbb{E}[n^2]}{\mathbb{E}[n]}$ is given by:

$$w_{\text{avg}} = \frac{\mathbf{s}^\top D[I - X(\mathbf{s})]^{-1} \mathbf{s}}{1 - g_{\text{node}}} + 1, \tag{24}$$

where $D = \text{diag}\{\mathbb{E}[k_1], \dots, \mathbb{E}[k_N]\}$ and $X(\mathbf{s})$ is a matrix function with the following elements

$$X_{i,j}(\mathbf{s}) = \frac{\mathbb{E}[(k_i k_j - \delta_{i,j} k_i)\mathbf{s}^{\mathbf{k}-\mathbf{e}_i-\mathbf{e}_j}]}{\mathbb{E}[k_i]}, \, i,j = 1, \dots, N. \tag{25}$$

When subjected to the bond percolation, $g_{\text{node}}$ and $w_{\text{avg}}$ become functions of the percolation parameter $p$.

$$g_{\text{node}}(p) = 1 - \mathbb{E}[(p(\mathbf{s}_p - 1) + 1)^{\mathbf{k}}], \tag{26}$$

where

$$(s_p)_i = \frac{\mathbb{E}\left[k_i(p(\mathbf{s}_p - 1) + 1)^{\mathbf{k}-\mathbf{e}_i}\right]}{\mathbb{E}[k_i]}, \, i = 1, \dots, N, \tag{27}$$

and

$$w_{\text{avg}}(p) = \frac{\mathbf{s}_p^\top D[p^{-1}I - X(p(\mathbf{s}_p - 1) + 1)]^{-1} \mathbf{s}_p}{1 - g_{\text{node}}(p)} + 1. \tag{28}$$

Physical systems that go through a phase transition, as spin glasses for example, feature a universal behaviour around critical points. Similar universality has also been shown to hold in critical networks during percolation[47]. We also notice that at $p = p_c$, when the network is critical, $w_{\text{avg}}(p)$ diverges to infinity. Moreover, as shown in the Methods section, when the giant

component emerges, this singularity is universally of type $\frac{1}{(p_c - p)}$:

$$\lim_{p \to p_c^-} \frac{w_{\text{avg}}(p)}{p_c - p} = \mathcal{O}(1). \qquad (29)$$

In the case of colour-dependent percolation, when $\mathbf{p}$ is a vector, it is enough to replace $p$ in Eqs (26)–(28) with $\text{diag}\{\mathbf{p}\}$:

$$g_{\text{node}}(\mathbf{p}) = 1 - \mathbb{E}[(\text{diag}\{\mathbf{p}\}(\mathbf{s}_p - 1) + 1)^{\mathbf{k}}], \qquad (30)$$

where

$$(s_p)_i = \frac{\mathbb{E}\left[k_i(\text{diag}\{\mathbf{p}\}(\mathbf{s}_p - 1) + 1)^{\mathbf{k} - \mathbf{e}_i}\right]}{\mathbb{E}[k_i]}, \; i = 1, \ldots, N, \qquad (31)$$

and

$$w_{\text{avg}}(\mathbf{p}) = \frac{\mathbf{s}_p^\top \mathbf{D}[\text{diag}\{\mathbf{p}\}^{-1}\mathbf{I} - \mathbf{X}(\text{diag}\{\mathbf{p}\}(\mathbf{s}_p - 1) + 1)]^{-1}\mathbf{s}_p}{1 - g_{\text{node}}(\mathbf{p})} + 1. \qquad (32)$$

We therefore see that the whole degree distribution contributes to the giant component size and its evolution during percolation. Equation (26) has been also derived in refs. [23,43] and (28) generalises a similar relation for unicoloured networks[16] to the case of arbitrary $N$. The Methods section presents derivations for Eqs. (21)–(32).

## Discussion

In this section, we compare the asymptotic theory against stochastic simulations on a few examples of generated networks. These examples are defined by simple analytical degree distributions for more transparent explanation. That said, the theory is also suitable and computationally feasible for any empirical degree distribution of arbitrary dimensionality. Consider a configuration model with three colours that is defined by

$$u(\mathbf{k}) = C \, \text{Poiss}(2k_1 + 3k_2 + 4k_3, 3), \; \mathbf{k} \neq 0, \qquad (33)$$

where $\text{Poiss}(k, \lambda) := e^{-\lambda}\frac{\lambda^k}{k!}$ is the Poisson mass density function and $C$ ensures the appropriate normalisation. This expression for the degree distribution was chosen since it features non-zero mixed moments, which in turn means that there are many nodes that bear edges of different colours. From Eqs. (3), (4) we obtain matrices $\boldsymbol{\mu}_0$, $\mathbf{M}$, $\mathbf{T}_1$, $\mathbf{T}_2$ and $\mathbf{T}_3$, which completely define the asymptotical properties of the finite components in the network. These matrices are then plugged into Eq. (6) to obtain the asymptote of the size distribution of connected components:

$$w_\infty(n) = 0.7992 n^{-3/2} e^{-0.00043n}. \qquad (34)$$

Figure 3a compares this asymptote to the size distribution obtained from simulations. In accordance with small theoretical value of $C_2 = 0.00043$, this asymptote appears to be almost scale-free. The network is very close to the critical point, however, one cannot assess if the network is just below or just above the phase transition by simply looking at the asymptote. Since $\mathbf{M}$ is a primitive matrix, we can safely apply the simplified test (18): the network is below its critical point since $\rho(\mathbf{M})^{-1} < 1$. As Fig. 3a illustrates, simple bond percolation with $p = 0.9$ and $p = 0.8$ gives a progressively faster decreasing size distribution.

The colour fractions in the finite components settle down on an uneven proportion. By applying Eq. (8), we find that

connected components have mean colour fractions as given by: $m_1 = 0.19$, $m_2 = 0.74$ and $m_3 = 0.7$. The covariance matrix of the colour fraction distribution (8), features negative pair correlations and vanishes when $n$ is large:

$$\boldsymbol{\Sigma} = \frac{1}{n} \begin{bmatrix} 0.22 & -0.20 & -0.02 \\ -0.20 & 0.25 & -0.05 \\ -0.02 & -0.05 & 0.07 \end{bmatrix}. \qquad (35)$$

The fact that covariance matrix vanishes for large components means that although the colour fractions fluctuate around $\mathbf{m}$, they converge to deterministic quantities in large components. This trend can also be seen in Fig. 3b that compares the theory with simulated colour fractions. The extent of the spread present in the scattered data in Fig. 3b is explained the theoretical covariance matrix given in Eq. (35) and therefore decreases as $\frac{1}{\sqrt{n}}$.

In the following example, we consider degree distribution

$$u(\mathbf{k}) = C \begin{cases} \text{Poiss}(k_1, 1.5), & k_2, k_3 = 0; \\ \text{Poiss}(k_2, 2.5), & k_1, k_3 = 0; \\ \text{Poiss}(k_3, 5), & k_1, k_2 = 0; \\ \alpha, & k_1, k_2, k_3 = 1. \end{cases} \qquad (36)$$

When $\alpha = 0$, each node has solely links of one colour, so that the whole system is a composition of three unicoloured networks. This fact results in a somewhat exotic situation when multiple giant components can stably coexist, which is an attractive phenomenon from the perspective of many applied disciplines[48]. When $\alpha > 0$, some nodes have links with different colours and the whole network contains, if any, one giant component. Nevertheless, as Fig. 4a shows, the network features high modularity in both cases, which points towards the utility of coloured edges for modelling networks with community structures. Conceptually, the difference between the cases of $\alpha > 0$ and $\alpha = 0$ is depicted by diagrams in respectively Fig. 2a, b.

In the case $\alpha = 0$, the dependence of $C_2$ on the percolation probability $p$, as given by the solid line in Fig. 4d, reveals that such criticalities occur three times at $p_c = \frac{1}{5}, \frac{2}{5}$, and $\frac{2}{3}$. Figure 4e shows that the weight-average component size is singular at these critical points. The values of $p_c$ can be easily deduced from matrix $\mathbf{M}$, which is diagonal when $\alpha = 0$:

$$\mathbf{M} = \begin{bmatrix} 2.5 & 0 & 0 \\ 0 & 1.5 & 0 \\ 0 & 0 & 5 \end{bmatrix}. \qquad (37)$$

In this case, the diagonal elements of $\mathbf{M}$ are also its eigenvalues, and all three eigenvectors are positive when normalised. So that, according to the criterion (12), all three eigenvalues are associated with valid phase transitions. Since all off-diagonal elements are zeros, one can invoke Eq. (20) to show that all secondary phase transitions coincide with the primary critical points. Note, that when $\alpha = 0$, matrix $\mathbf{M}$ not primitive, and therefore the simplified phase transition criterion (18) is not applicable.

Setting $\alpha = 0.1$ in degree distribution (36) perturbs $\mathbf{M}$ so that it becomes a full matrix with small off-diagonal elements,

$$\mathbf{M} = \begin{bmatrix} 2.4 & 0.1 & 0.02 \\ 0.04 & 1.4 & 0.02 \\ 0.04 & 0.1 & 4.71 \end{bmatrix}. \qquad (38)$$

Although the total size of the giant components, as shown in Fig. 4e, f differs only a little, the eigenvalue decomposition of

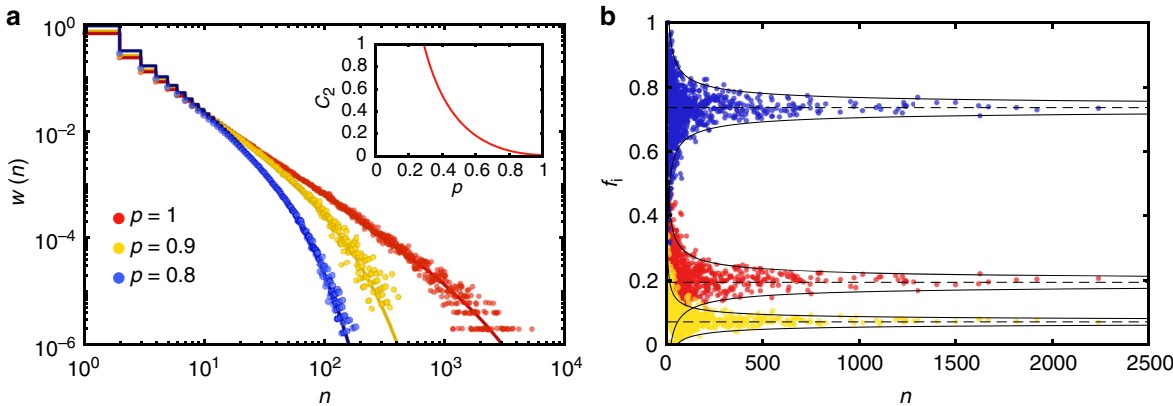

**Fig. 3** Simple bond percolation in a sub-critical coloured network. **a** Simulated size distributions of connected components (scatter plots) are compared with the theoretical asymptotes (solid lines). Inset: the dependence of the exponent coefficient $C_2$ on the percolation probability $p$. **b** Fraction of coloured edges $f_i$ that belong to components of size $n$. Scatter plots: simulated networks (1-red, 2-blue, 3-yellow). Dashed lines: theoretical mean values, $m_i$. Solid lines: theoretical $2\sigma$ deviation is vanishing as $\frac{1}{\sqrt{n}}$

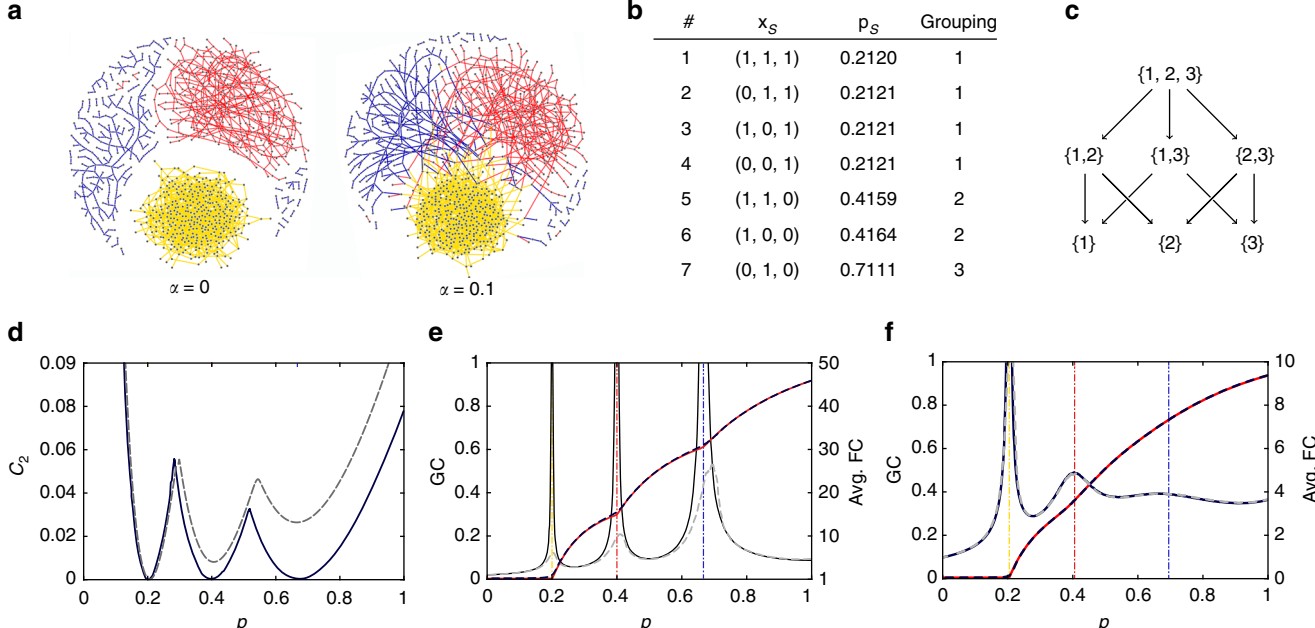

**Fig. 4** Multiple phase transitions during percolation in the edge-coloured network model. **a** Samples of network topologies corresponding to the degree distribution given by Eq. (36) with $\alpha = 0$ and $\alpha = 0.1$. **b** The list of all secondary phase transitions and the colour subset indicators that correspond to them. **c** The partial order of all secondary phase transitions in a network with three colours. Arrows denote inclusion of the colour subsets. **d** Dependance of the exponent coefficient $C_2$ on the percolation parameter $p$ for two vales of the parameter: $\alpha = 0$ (solid line), and $\alpha = 0.1$ (dashed line). **e, f** Left axis: The total size of all giant components as a function of percolation parameter $p$ as obtained from the theory (red solid line) and simulations (black dashed line). Right axis: The weight-average size of finite connected components as obtained form the theory (black solid line) and simulations (grey dashed line). These results are obtained for degree distributions with: **e** $\alpha = 0$, and **f** $\alpha = 0.1$. The vertical guide lines indicate theoretical values for phase transition points

$M$ shows that there is only one positive eigenvector, and therefore, one phase transition. This observation is supported by the fact that $C_2(p) = 0$ only once, as shown by the dashed line in Fig. 4d, and that the size average of connected component, as defined by Eq. (28), is singular only at $p \approx 0.21$ (see Fig. 4f). One can yet observe that $C_2(p)$ has two local minima at the locations where the case of $\alpha = 0$ features phase transitions. These local minima correspond to points where the average component size has maxima: a very similar phenomenon of "multiple phase transitions" was also observed in empirical data[9]. This constitutes an intricate situation: on one hand we know that Eq. (12) has multiple solutions only when $M$

is not a primitive matrix, which corresponds to a disjoint network. On the other hand, when a fully connected network has strongly segregated communities, $C_2(p)$ may drop multiple times so low that one is not able to distinguish the empirical size distribution form being not scale-free. See also ref. [26] for a similar discussion. Moreover, since $M$ is strongly diagonal dominant for $\alpha = 0.1$, the secondary critical points appear in groups. The full list, as obtained from criterion (19), is given in Fig. 4b. Phase transitions associated with three colours feature the hierarchy as indicated by the partially ordered set in Fig. 4c, one may also think of Fig. 4b as a linear sorting of this partial order.

Probably the most surprising implication of the equation describing the internal structure of connected components is not that the colour fractions settle on an uneven ratio, as depicted in Fig. 3b, but that this ratio peculiarly evolves as a function of $p$. For instance, in the case of $\alpha = 0.1$, the colour fractions $f_1$, $f_2$ and $f_3$ feature a switching behaviour. None of the switching points coincides with the phase transitions, but as Fig. 5a reveals, they are rather equidistant from the critical points. This trend may be exploited to device early warning strategies that, similar to those devised in ref. [39], detect proximity of a phase transition in empirical networks. In contrast to large components, in which the colour fractions converge to a deterministic values as shown in Fig. 3b, the structure of small components is not deterministic but features fluctuations. The spread of these fluctuations (as given by $\mathbf{\Sigma}$) is predicted by the theory and if the component size is fixed, this spread does not vanish even when the total size of the network is infinitely large. Figure 5c illustrates how the spread of simulated data points in a small component ($n = 50$) follows the theoretical predictions as given by this covariance matrix. Although the colour switching during percolation can be predicted theoretically, it is hard to say what are the sufficient prerequisite for this phenomenon to occur. Supplementary Table 1 provides a parametric study of the colour switching profiles in networks with various proportions of nodes that bear edges of different colours. The latter quantity is a proxy for how modular is the test network. On the basis of this observation we conjecture that segregated networks with large difference in community sizes are more prone to colour switching. Another observation one can derive from Fig. 5b, is that the mean colour fraction in finite components is often drastically different from that in the giant component (see also Supplementary Table 1). Such difference on the level of nodes of these large structures may pave way to differentiating if a small sample of nodes was taken from the giant component or from a finite one.

In colour-dependent percolation, one investigates the properties of the percolated network as a function of probability vector $\mathbf{p} = (p_1, p_2\ p_3)$. All configurations of $\mathbf{p}$ amount to the volume of a unit cube. Figure 6 presents the regions of this parameter space where the network becomes critical, that is $C_2 = 0$, or close-to-critical, $C_2$ is small. This configurations are recovered by numerically solving Eq. (16) that parametrises the corresponding manifolds. One can see that what appears as single curves in Fig. 4d, e for simple percolation, is now a surface placed in the unit cube, see Fig. 6. When $p_1 = p_2 = p_3$, which corresponds to a diagonal of the cube as indicated by the yellow line, the colour-dependent and simple percolations are equivalent. Figure 6b shows that in the case $\alpha = 0.1$, the critical points form a box-shaped surface, whereas $\alpha = 0$ changes this surface into a more complex shape, see Fig. 6a. Note, in Fig. 6a the yellow line intersects this surface three times, which corresponds to the three phase-transition points that are also observed in Fig. 4e. Although there is a conceptual difference between the cases presented in Fig. 6a, b, one would expect that the actual networks should be close in some sense. Indeed, Fig. 6b is obtained by perturbing the network represented in Fig. 6a with a small parameter $\alpha$. This similarity can be highlighted if we compare the isosurfaces at which $C_2$ is small. In Fig. 6c, d: one can see that these isosurfaces do bear resemblance.

The time evolution of the colour-dependent percolation can be represented with a path $\mathbf{p}(t)$, $t \in [0, 1]$ that starts at $\mathbf{p}(1) = (1, 1, 1)^{\top}$, which corresponds to the intact network, and ends at $\mathbf{p}(0) = (0, 0, 0)^{\top}$—a completely disintegrated one. (To keep the analogy with the simple percolation we let time to go in reverse.) Furthermore, the question of how an individual network responds to percolation is about how this path relates to

the above-described geometric structures, and it turns out that colour-dependent percolation can lead to completely different types of behaviour that are not observed in simple percolation. In order to demonstrate this fact, we devise a path $\mathbf{p}(t)$ in such a way that $C_2(\mathbf{p})$ is minimal on it. This is achieved by solving the path minimisation: $\int_0^1 C_2(\mathbf{p}(t))\mathrm{d}t \to \min$. Our target is thus to reproduce the scenario given in Fig. 2c. The optimal paths for the networks with $\alpha = 0$ and $\alpha = 0.1$ are illustrated in Fig. 7a, d, whereas Fig. 7b, e depict the corresponding profiles of $C_2(t)$. One immediately notices that in these examples the profiles of $C_2(t)$ vanish not at discrete points, as was the case with simple percolation, but on continuous intervals. Everywhere on this intervals the network is critical: that is the sizes of connected components feature the scale-free behaviour and thus the weight-average component size is infinite. As Fig. 7c, f shows, this theoretical considerations are also supported by numerically generated networks. Generated networks comprise of finite number of nodes, and therefore, they cannot feature infinite average component size. However, this quantity features macroscopic fluctuations within the critical interval and diverges to infinity with growing system size. Remarkably, the width of this critical interval does not shrink to zero in infinite systems. This phenomenon constitutes a previously undocumented type of phase transitions in coloured networks—the phase transitions with wide critical regimes. Within this critical regimes, the average size of a connected component features macroscopic fluctuations. Such a behaviour is somewhat reminiscent to critical windows found in finite random graphs[49,50].

To summarise, in this paper edge colours provide an abstraction of an additional layer of information that a network can be equipped with, as well as an abstraction of various network structures. The colours may represent an affiliation to communities, multiplexity, different types of interactions, assortative/disassortative relationships and other aspects prevalent to complex networks. We have shown that colour dependencies may amplify failures in connectivity or make the collapse of the network less sudden. This is why this theory provides the foundation for answering the key question of complex networks science: "How can we economically design robust multilayer networks of infrastructures or financial networks with trustworthy links?"[51]. In many ways a modeller may exploit the above-described geometric interpretation of colour-dependent percolation in sake of network design and control. By following similar motivations to the ones in ref. [4], one could aim to optimise the percolation path so that minimal/maximal number of edges is removed before percolation reaches the phase transition, or for a fixed path, one might optimise the network robustness so that the critical manifold avoids the path as much as possible. Another objective, reducing the sharpness of the phase transition[35,52], can be achieved by reducing the angle between the path and the surface at the intersection point. In fact, as we have demonstrated by an example, one may even construct a path that does not immediately intersect the manifold but stays inside for a long time, and therefore, keeps the network in the critical regime. The latter observation shifts the paradigm of the critical point itself, as it demonstrates that even in infinite systems, the criticality may occur continuously on a whole, non-vanishing interval.

## Methods

**Summary of the asymptotic theory for unicoloured networks**. When there is only one type of edges, $N = 1$, $u(k)$ is simply the probability that a randomly selected node bears $k$ edges. In the configuration model, the size distribution of

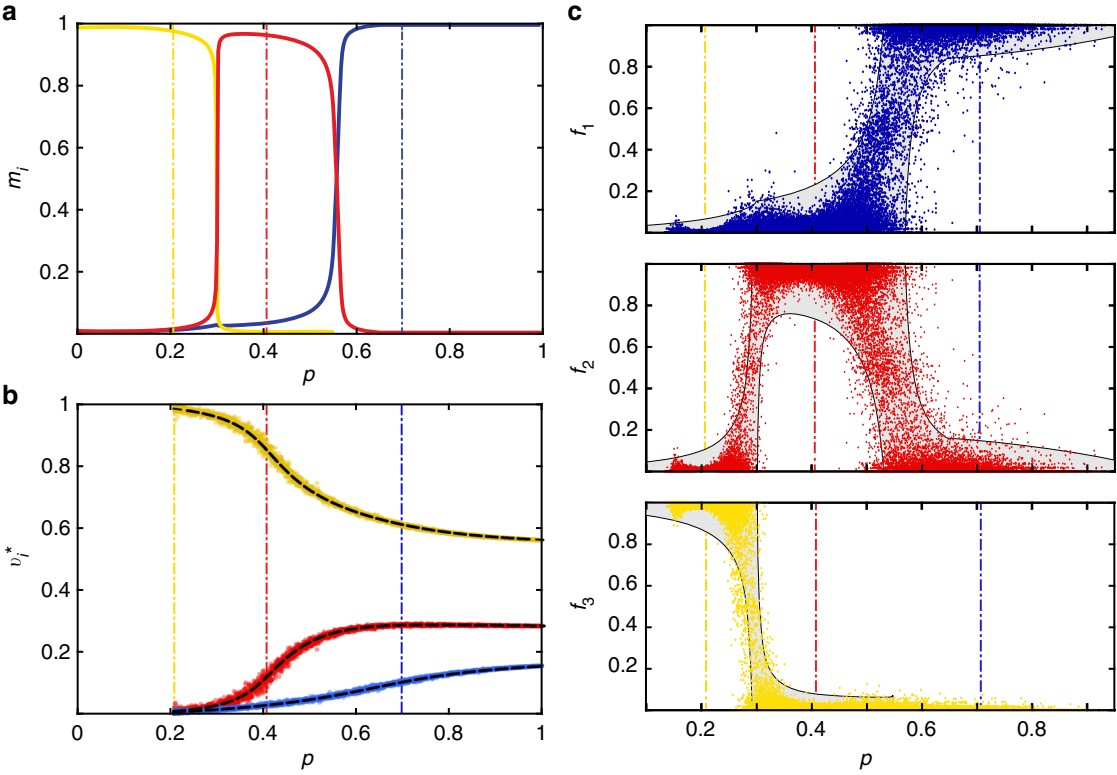

**Fig. 5** Colour switching in finite components. **a** Theoretical expected fraction of colours $m_i$ as a function of percolation probability $p$ features a switching behaviour in the example network with $\alpha = 0.1$. These profiles are the same for all finite components independently of their size. **b** Theoretical and observed fraction of colours in the giant component. **c** The $2\sigma$ spread of random variables $f_1$, $f_2$ and $f_3$ as predicted in the components of size $n = 50$ is compared to observations in generated networks. The vertical guide lines mark values of $p$ at which the disjoint network features phase transitions. A parametric study describing how colour fractions evolve in networks with various modularity is given in Supplementary Table 1

connected components is given by convolution powers of the degree distribution[30]:

$$w(n) = \begin{cases} \frac{\mathbb{E}[k]}{n-1} u_1^{*n}(n-2), & n > 1, \\ u(0) & n = 1. \end{cases} \quad (39)$$

Here $u_1(k) = \frac{(k+1)u(k+1)}{\mathbb{E}[k]}$ is the excess degree distribution[29], the convolution product $d(n) = f(n) * g(n)$ is defined as

$$d(n) := \sum_{j+k=n} f(j)g(k), \; j, k \geq 0, \quad (40)$$

and the convolution power is defined by induction: $f(k)^{*n} = f(k)^{*n-1} * f(k)$, $f(k)^{*0} := \delta(k)$. When the degree distribution has a light tail, the Eq. (39) features the universal asymptote[30],

$$w_\infty(n) = C_1 e^{-C_2 n} n^{-3/2}. \quad (41)$$

The expressions for $C_1$ and $C_2$ are given in terms of the first three moments: $\mathbb{E}[k]$, $\mathbb{E}[k^2]$, and $\mathbb{E}[k^3]$. Namely, $C_1 = \frac{\mathbb{E}[k]^2}{\sqrt{2\pi(\mathbb{E}[k]\mathbb{E}[k^3]-\mathbb{E}[k^2]^2)}}$, $C_2 = \frac{(\mathbb{E}[k^2]-2\mathbb{E}[k])^2}{2(\mathbb{E}[k]\mathbb{E}[k^3]-\mathbb{E}[k^2]^2)}$. At the critical point, when $C_2 = 0$, the asymptote (41) indicates scale-free behaviour, and features exponent $-\frac{3}{2}$. The condition $C_2 = 0$ is equivalent to Molloy and Reed giant component criterion:

$$\mathbb{E}[k^2] - 2\mathbb{E}[k] = 0. \quad (42)$$

The next section derives a generalisation of this theory for the case of arbitrary number of colours, in which case the coefficients of the N-colour asymptote are derived in terms of mixed moments up to the third order.

**Size distribution of connected components**. In a network with $N$ colours, the degree distribution $u(\mathbf{k})$, $\mathbf{k} = (k_1, \ldots, k_N)$ is the probability that a randomly selected node bears $k_i$ edges of colour $i = 1, 2, \ldots, N$. Let $\boldsymbol{\mu}_0$ is the vector-valued expectation

of this distribution,

$$\boldsymbol{\mu}_0 = (\mathbb{E}[k_1], \mathbb{E}[k_2], \ldots, \mathbb{E}[k_N])^\top. \quad (43)$$

If one choses a node at the end of a random, $i$-coloured edge instead of choosing a node at random, the corresponding degree distribution is called $i$-excess:

$$u_i(\mathbf{k}) = (\mathbf{k} + \mathbf{e}_i) \frac{u(\mathbf{k} + \mathbf{e}_i)}{\mathbb{E}[k_i]}, \quad (44)$$

where $\mathbf{e}_i$ are the standard basis vectors. The expected column-vectors $\boldsymbol{\mu}_i$ of $i$-excess degree distributions can be expressed in terms of expectations of $u(\mathbf{k})$:

$$(\boldsymbol{\mu}_i)_j = \frac{\mathbb{E}[k_i k_j]}{\mathbb{E}[k_i]} - \delta_{i,j}, \; i, j = 1, \ldots, N. \quad (45)$$

For convenience of notation, we introduce $\mathbf{M} = (\boldsymbol{\mu}_1, \boldsymbol{\mu}_2, \ldots, \boldsymbol{\mu}_N)$, a matrix that contains $\boldsymbol{\mu}_i$ as its columns. The covariance matrices of $i$-excess degree distributions are given by

$$(\mathbf{T}_i)_{j,l} = \frac{\mathbb{E}[k_i k_j k_l]}{\mathbb{E}[k_i]} - \frac{\mathbb{E}[k_i k_j]\mathbb{E}[k_i k_l]}{\mathbb{E}[k_i]^2}, \; i, j, l = 1, \ldots, N. \quad (46)$$

In a similar fashion to how Eq. (39) was derived, ref. [33] applies Joyal's theory of combinatorial species[53] to write the exact expression for the size distribution of weakly connected components:

$$w(n) = \sum_{\substack{k_1 + \ldots + k_N = n-1 \\ k_i \geq 0}} \left( \widetilde{D} * u * u_1^{*k_1} * \ldots * u_N^{*k_N} \right)(\mathbf{k}), \quad (47)$$

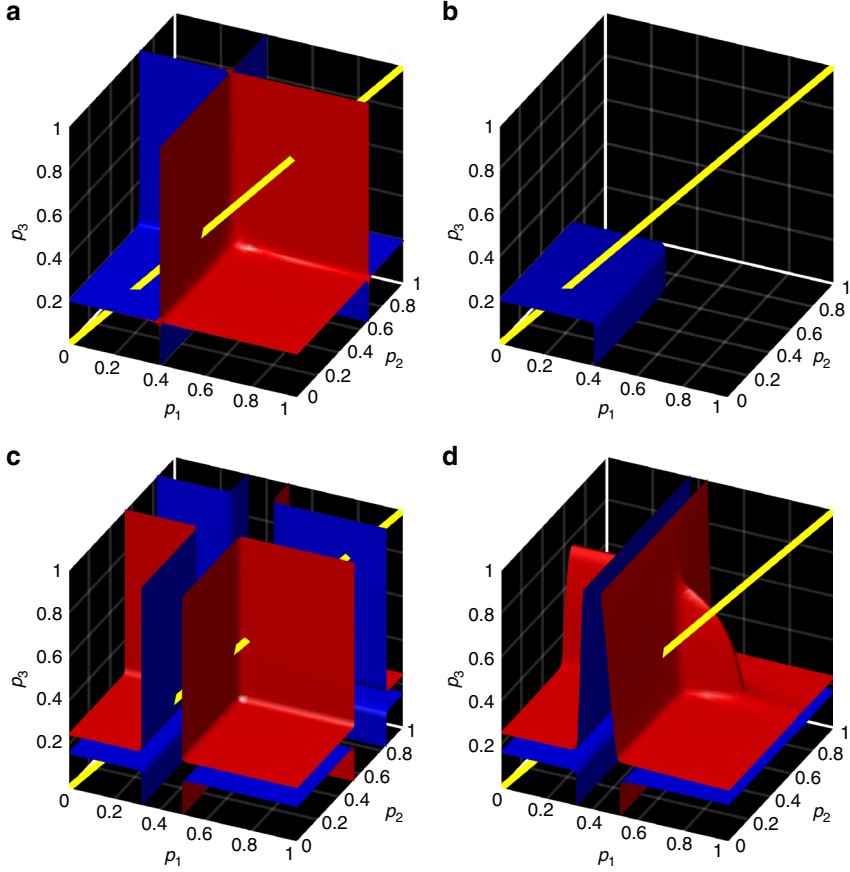

**Fig. 6** Colour-dependent percolation in a network with three colours. **a**, **b** Surfaces containing critical configurations of vector **p** (critical manifolds) in the network model with: **a** $\alpha = 0$ and **b** $\alpha = 0.1$. **c**, **d** Manifolds containing **p**-configurations for which $C_2$ is small (isosurface $C_2 = 0.02$) are computed for: **c** $\alpha = 0$ and **d**. $\alpha = 0.1$. The red-blue colouring is used for better visual distinction between the surface's sides in **a**, **b** and manifold's interior/exterior in **c**, **d**. The diagonal yellow line indicates the configurations $p_1 = p_2 = p_3$ for which the problem degenerates to the simple bond percolation

where the $N$-dimensional convolution $d(\mathbf{n}) = f(\mathbf{n}) * g(\mathbf{n})$, is defined as

$$d(\mathbf{n}) := \sum_{\mathbf{j}+\mathbf{k}=\mathbf{n}} f(\mathbf{j})g(\mathbf{k}), \ \mathbf{j}, \mathbf{k} \geq 0. \tag{48}$$

The sum in Eq. (48) runs over all partitions of the $N$-dimensional vector $\mathbf{n}$ into two non-negative terms $\mathbf{j}$ and $\mathbf{k}$; $\widetilde{D} = \det_*(D)$ refers to the determinant computed with the multiplication replaced by the convolution, and matrix $D$ has the following elements:

$$D_{i,j} = \delta(\mathbf{k})\delta_{i,j} - [k_j u_i(\mathbf{k})] * u_i(\mathbf{k})^{*(-1)}, \ i,j = 1, \ldots, N. \tag{49}$$

**The asymptote of the size distribution**. The size distribution given in Eq. (47) features the following asymptote when $n$ is large:

$$w_\infty(n) = C_1 n^{-3/2} e^{-C_2 n}. \tag{50}$$

This asymptote has to coefficients $C_1 > 0$ and $C_2 > 0$. We first define the exponent coefficient $C_2$:

$$C_2 = \frac{1}{2}[1, \mathbf{z}^{*\mathrm{T}}]K^{\mathrm{T}}\sigma_{\mathbf{z}^*}^{-1}K\begin{bmatrix}1\\\mathbf{z}^*\end{bmatrix}, \tag{51}$$

where

$$\sigma_{\mathbf{z}} = (1 - |\mathbf{z}|)T_1 + \sum_{i=1}^{N-1} z_i T_{i+1}, \ |\mathbf{z}| := \sum_{i=1}^{N-1} z_i, \tag{52}$$

and $K = (M - I)\mathbf{A}^{-1}$,

$$A_{i,j} = \begin{cases} 1 & i = 1, \\ \delta_{i,j} & i > 1, \end{cases} \tag{53}$$

and $\mathbf{z}^* = \arg\min f(\mathbf{z})$, $\mathbf{z}^* > 0$, $|\mathbf{z}^*| \leq 1$, is the point where $f(\mathbf{z})$ reaches its minimum value:

$$f(\mathbf{z}) := \frac{1}{2}[1, \mathbf{z}^{\top}]K^{\top}\sigma_{\mathbf{z}}^{-1}K\begin{bmatrix}1\\\mathbf{z}\end{bmatrix}, \tag{54}$$

Note that $\mathbf{z}$ is a vector of dimension $N - 1$.
The scaling coefficient $C_1$ is given by:

$$C_1 = \det(M - I)\mathrm{tr}(\mathrm{adj}(Q)R)\left(\frac{\det[\mathbf{H}_{\mathbf{z}^*}^{-1}]}{2\pi\det[\sigma_{\mathbf{z}^*}]}\right)^{1/2} e^{-[1,\mathbf{z}^{*\top}]K^{\top}\sigma_{\mathbf{z}^*}^{-1}\boldsymbol{\mu}_0}, \tag{55}$$

where $Q_{i,j} = \delta_{i,j} - [1, \mathbf{z}^*]^{\top}K^{\top}\sigma_{\mathbf{z}^*}^{-1}\mathrm{adj}(M - I)T_i \mathbf{e}_j$, and $R_{i,j} = \boldsymbol{\mu}_0^{\top}\sigma_{\mathbf{z}^*}^{-1}\mathrm{adj}(M - I)T_i \mathbf{e}_j$ are square matrices of size $N \times N$, and

$$(\mathbf{H}_{\mathbf{z}})_{i,j} = [1, \mathbf{z}^{\top}]K^{\top}\sigma_{\mathbf{z}}^{-1}(T_{j+1} - T_1)\sigma_{\mathbf{z}}^{-1}(T_{i+1} - T_1)\sigma_{\mathbf{z}}^{-1}K\begin{bmatrix}1\\\mathbf{z}\end{bmatrix} + \mathbf{e}_{i+1}^{\top}K^{\top}\sigma_{\mathbf{z}}^{-1}K\mathbf{e}_{j+1}$$
$$- [1, \mathbf{z}^{\top}]K^{\top}\sigma_{\mathbf{z}}^{-1}(T_{j+1} - T_1)\sigma_{\mathbf{z}}^{-1}K\mathbf{e}_{i+1} - [1, \mathbf{z}^{\top}]K^{\top}\sigma_{\mathbf{z}}^{-1}(T_{i+1} - T_1)\sigma_{\mathbf{z}}^{-1}K\mathbf{e}_{j+1} \tag{56}$$

is the Hessian matrix of $f(\mathbf{z})$ having size $N - 1$ by $N - 1$, the standard basis vectors are denoted by $(\mathbf{e}_i)_j = \delta_{i,j}$. The next section derives this expressions.

**Derivations for the asymptote of the component size distribution**. This section derives the asymptote for the Eq. (47). Note that in Eq. (47), the arguments of the convolved functions and the convolution powers are both taken from the same vector $\mathbf{k}$. In order to circumvent this issue, let us decouple the argument from the powers. We thus define $G(\mathbf{k}', \mathbf{k}) := u_1(\mathbf{k}')^{*k_1} * \ldots * u_N(\mathbf{k}')^{*k_N}$. To introduce the

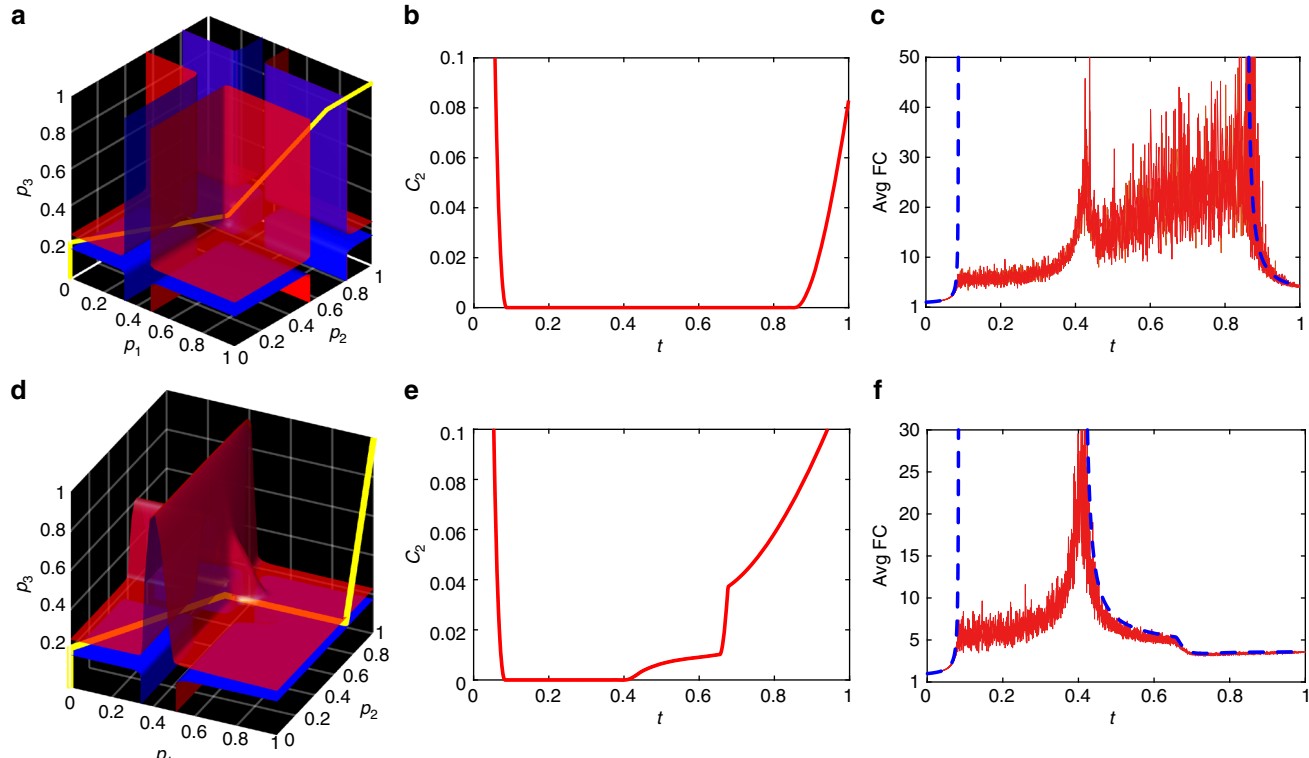

**Fig. 7** Colour-dependent percolation that features wide critical regimes. Networks corresponding to the degree distribution defined by Eq. (36) with: **a–c** $\alpha = 0$ and **d–f** $\alpha = 0.1$. **a**, **d** Optimal percolation paths $\mathbf{p}(t)$ together with the reference isosurfaces ($C_2 = 0.02$). **b**, **e** The $C_2$-profiles corresponding to the optimal paths. **c**, **f** Comparisons of the theoretical weighted-average component size profiles against the corresponding profiles extracted from stochastically generated networks with $10^5$ nodes

idea of the asymptotic analysis, we first start with a simplified version of (47).

$$w'(n+1) = \sum_{\substack{k_1+\ldots+k_N=n \\ k_i \geq 0}} u_1(\mathbf{k})^{*k_1} * \ldots * u_N(\mathbf{k})^{*k_N} = \sum_{\substack{k_1+\ldots+k_N=n \\ k_i \geq 0}} G(\mathbf{k},\mathbf{k}). \quad (57)$$

Thus we will now study the asymptotical behaviour of the product of convolution powers. Let $\phi(\boldsymbol{\omega})$ denotes the characteristic function of $u(\mathbf{k})$,

$$\phi(\boldsymbol{\omega}) = \sum_{\mathbf{k} \geq 0} e^{i\boldsymbol{\omega}^\top \mathbf{k}} u(\mathbf{k}), \; \boldsymbol{\omega} \in \mathbb{R}^N, \, i^2 = -1, \quad (58)$$

and $\phi_i(\boldsymbol{\omega})$ denote the characteristic functions of $u_i(\mathbf{k})$. Then, by keeping the second argument of $G(\mathbf{k}', \mathbf{k})$ as a parameter, the characteristic function is given by the product:

$$\psi(\boldsymbol{\omega}, \mathbf{k}) = \phi(\boldsymbol{\omega}) \prod_{i=1}^N \phi_i^{k_i}(\boldsymbol{\omega}), \quad (59)$$

however, as follows from the central limit theorem, $\phi_i^{k_i}(\boldsymbol{\omega})$ approach the limit functions when $k_i$ is large:

$$\lim_{k_i \to \infty} |\phi_i^{k_i}(\boldsymbol{\omega}) - \phi_{i,\infty}^{k_i}(\boldsymbol{\omega})| = 0, \quad (60)$$

where

$$\phi_{i,\infty}^{k_i}(\boldsymbol{\omega}) = e^{ik_i \boldsymbol{\mu}_i^\top \boldsymbol{\omega} - \frac{1}{2}\boldsymbol{\omega}^\top k_i T_i \boldsymbol{\omega}}, \quad (61)$$

and $\boldsymbol{\mu}_i$ and $T_i$ are the expected values and covariance matrices as defined by (45) and (46). Applying similar procedure to the Eq. (59), we obtain:

$$\lim_{k_1,\ldots,k_N \to \infty} |\psi(\boldsymbol{\omega}, \mathbf{k}) - \psi_\infty(\boldsymbol{\omega}, \mathbf{k})| = 0, \quad (62)$$

where

$$\psi_\infty(\boldsymbol{\omega}, \mathbf{k}) = e^{i\boldsymbol{\omega}(M\mathbf{k}+\boldsymbol{\mu}_0) - \frac{1}{2}\boldsymbol{\omega}^\top \sigma(k)\boldsymbol{\omega}}, \quad (63)$$

and $\sigma(\mathbf{k}) = \sum_{i=1}^N k_i T_i + o(\mathbf{k})$. Inverse flourier transform of Eq. (63) is easy to obtain since $\psi_\infty(\boldsymbol{\omega}, \mathbf{k})$ itself is the characteristic function of the multivariate Gaussian function:

$$G(\mathbf{k}', \mathbf{k}) = \det[2\pi\sigma(\mathbf{k})]^{-1/2} e^{-\frac{1}{2}(\mathbf{k}'-M\mathbf{k}-\boldsymbol{\mu}_0)^\top \sigma^{-1}(\mathbf{k})(\mathbf{k}'-M\mathbf{k}-\boldsymbol{\mu}_0)}$$
$$= \det[2\pi(\mathbf{k})]^{-1/2} e^{-(M\mathbf{k}-\mathbf{k}')^\top \sigma^{-1}(\mathbf{k})\boldsymbol{\mu}_0 - \frac{1}{2}\boldsymbol{\mu}_0^\top \sigma^{-1}(\mathbf{k})\boldsymbol{\mu}_0} e^{-\frac{1}{2}(\mathbf{k}'-M\mathbf{k})^\top \sigma^{-1}(\mathbf{k})(\mathbf{k}'-M\mathbf{k})}. \quad (64)$$

Consider the following matrix,

$$(\mathbf{A})_{i,j} = \begin{cases} 1 & i=1, \\ \delta_{i,j} & i>1, \end{cases} \quad (65)$$

and the transformation it induces, $\mathbf{z}' = \frac{1}{n}\mathbf{A}\mathbf{k}$. Since $k_1 + k_2 + \ldots + k_N = n$, we have $z'_1 = 1$ and therefore we may write $\mathbf{z}' = \begin{bmatrix} 1 \\ \mathbf{z} \end{bmatrix}$, where $\mathbf{z}$ is a vector of dimension $N-1$. Let us now set $\mathbf{k}' = \mathbf{k}$ and introduce the transformation of variables $\mathbf{k} = n\mathbf{A}^{-1}\begin{bmatrix} 1 \\ \mathbf{z} \end{bmatrix}$, which replaces the functions appearing in the exponent of (64) with

$$\frac{1}{2}(\mathbf{k}'-M\mathbf{k})^\top \sigma^{-1}(\mathbf{k})(\mathbf{k}'-M\mathbf{k}) = \frac{1}{2}n[1, \mathbf{z}^\top]K^\top \sigma_\mathbf{z}^{-1} K \begin{bmatrix} 1 \\ \mathbf{z} \end{bmatrix} = nf(\mathbf{z}), \quad (66)$$

and

$$(M\mathbf{k}-\mathbf{k}')^\top \sigma^{-1}(\mathbf{k})\boldsymbol{\mu}_0 + \frac{1}{2}\boldsymbol{\mu}_0^\top \sigma^{-1}(\mathbf{k})\boldsymbol{\mu}_0 = [1, \mathbf{z}^\top]K^\top \sigma_\mathbf{z}^{-1}\boldsymbol{\mu}_0 + \frac{1}{2n}\boldsymbol{\mu}_0^\top \sigma_\mathbf{z}^{-1}\boldsymbol{\mu}_0$$
$$= g(\mathbf{z}) + O(n^{-1}), \quad (67)$$

where $g(\mathbf{z}) = [1, \mathbf{z}^\top] \mathbf{K}^\top \sigma_\mathbf{z}^{-1} \boldsymbol{\mu}_0$, $\mathbf{K} = (\mathbf{M} - \mathbf{I})\mathbf{A}^{-1}$ and

$$\sigma_\mathbf{z} = (1 - |\mathbf{z}|)\mathbf{T}_1 + \sum_{i=1}^{N-1} z_i \mathbf{T}_{i+1}, \quad |\mathbf{z}| := \sum_{i=1}^{N-1} z_i, \qquad (68)$$

are independent of $n$. The Taylor expansion of $f(\mathbf{z})$ around the point where this function reaches its minimum value,

$$\mathbf{z}^* = \arg\min f(\mathbf{z}), \qquad (69)$$

gives:

$$f(\mathbf{z}) = f(\mathbf{z}^*) + \nabla f(\mathbf{z}^*)^\top (\mathbf{z} - \mathbf{z}^*) + \frac{1}{2}(\mathbf{z} - \mathbf{z}^*)^\top \mathbf{H}_{\mathbf{z}^*}(\mathbf{z} - \mathbf{z}^*) + \mathbf{R}(\mathbf{z}, \mathbf{z}^*), \qquad (70)$$

where the $N-1$-by-$N-1$ matrix $\mathbf{H}_{\mathbf{z}^*}$ is the Hessian of $f(\mathbf{z})$ at $\mathbf{z} = \mathbf{z}^*$:

$$(\mathbf{H_z})_{i,j} = [1, \mathbf{z}^\top]\mathbf{K}^\top \sigma_\mathbf{z}^{-1}(\mathbf{T}_{j+1} - \mathbf{T}_1)\sigma_\mathbf{z}^{-1}(\mathbf{T}_{i+1} - \mathbf{T}_1)\sigma_\mathbf{z}^{-1}\mathbf{K}\begin{bmatrix}1\\\mathbf{z}\end{bmatrix} + \mathbf{e}_{i+1}^\top \mathbf{K}^\top \sigma_\mathbf{z}^{-1}\mathbf{K}\mathbf{e}_{j+1}$$
$$-[1, \mathbf{z}^\top]\mathbf{K}^\top \sigma_\mathbf{z}^{-1}(\mathbf{T}_{j+1} - \mathbf{T}_1)\sigma_\mathbf{z}^{-1}\mathbf{K}\mathbf{e}_{i+1} - [1, \mathbf{z}^\top]\mathbf{K}^\top \sigma_\mathbf{z}^{-1}(\mathbf{T}_{i+1} - \mathbf{T}_1)\sigma_\mathbf{z}^{-1}\mathbf{K}\mathbf{e}_{j+1}, \qquad (71)$$

and $\mathbf{R}(\mathbf{z}, \mathbf{z}^*)$ is the expansion residue. Note that the gradient $\nabla f(\mathbf{z}^*) = 0$ since $\mathbf{z}^*$ is a minimum of $f(\mathbf{z})$, and by plugging these expansion into (64) we obtain:

$$F_n(\mathbf{z}) := G(\mathbf{k}, \mathbf{k}) = \det[2\pi\sigma_\mathbf{z}]^{-1/2} e^{-g(\mathbf{z})} e^{-nf(\mathbf{z})}$$
$$= \det[2\pi\sigma_\mathbf{z}]^{-1/2} e^{-g(\mathbf{z})} e^{-nf(\mathbf{z}^*) - \frac{1}{2}(\mathbf{z}-\mathbf{z}^*)^\top n\mathbf{H}_{\mathbf{z}^*}(\mathbf{z}-\mathbf{z}^*) - n\mathbf{R}(\mathbf{z},\mathbf{z}^*)}. \qquad (72)$$

Now, by multiplying $F_n(\mathbf{z})$ with $1 = \frac{\det[2\pi\frac{1}{n}\mathbf{H}_{\mathbf{z}^*}^{-1}]^{1/2}}{\det[2\pi\frac{1}{n}\mathbf{H}_{\mathbf{z}^*}^{-1}]^{1/2}}$ and rewriting the sum of exponents as a product of exponential functions isolates a multivariate Gaussian function in the expression of $F_n(\mathbf{z})$:

$$F_n(\mathbf{z}) = \frac{e^{-\frac{1}{2}(\mathbf{z}-\mathbf{z}^*)^\top[\frac{1}{n}\mathbf{H}_{\mathbf{z}^*}^{-1}]^{-1}(\mathbf{z}-\mathbf{z}^*)}}{\det[2\pi\frac{1}{n}\mathbf{H}_{\mathbf{z}^*}^{-1}]^{1/2}} \frac{\det[2\pi\frac{1}{n}\mathbf{H}_{\mathbf{z}^*}^{-1}]^{1/2}}{\det[2\pi\sigma_\mathbf{z}]^{1/2}} e^{-g(\mathbf{z})} e^{-\frac{1}{2}nf(\mathbf{z}^*) - \frac{1}{2}n\mathbf{R}(\mathbf{z},\mathbf{z}^*)}. \qquad (73)$$

Let us turn back to the asymptotic analysis of the Eq. (57), which becomes now written as:

$$w'(n+1) = \sum_{\substack{k_1 + \ldots + k_N = n \\ k_i \geq 0}} F_n\left(\frac{1}{n}\mathbf{A}\mathbf{k}\right) = \sum_{\mathbf{z} \in \Omega_n} F_n(\mathbf{z}). \qquad (74)$$

The latter summation is performed over a sequence of sets:

$$\Omega_n := \left\{(1, z_1, \ldots, z_{N-1})^\top : |\mathbf{z}| \leq 1, \ z_i = \frac{m}{n}, \ m = 0, \ldots, n\right\}, \qquad (75)$$

which, as $n \to \infty$, becomes dense in the limiting set

$$\Omega_\infty := \{(1, z_1, \ldots, z_{N-1}) : |\mathbf{z}| \leq 1, \ z_i > 0, \ z_i \in \mathbb{R}\}, \qquad (76)$$

so that, on one hand, the sum form (74) converges to the integral

$$\lim_{n \to \infty} \left| \sum_{\mathbf{z} \in \Omega_n} F_n(\mathbf{z}) - n^{N-1} \int_{\Omega_\infty} F_n(\mathbf{z})d\mathbf{z} \right| = 0, \qquad (77)$$

and on another hand, the first fraction in Eq. (73) is a properly normalised multivariate Gaussian function with mean $\mathbf{z}^*$ and shrinking variance $\frac{1}{n}\mathbf{H}_{\mathbf{z}^*}^{-1}$. This Gaussian function approaches the Dirac's delta $\delta(\mathbf{z} - \mathbf{z}^*)$ in the large $n$ limit: its variance vanishes as $O(\frac{1}{n})$ while the mean remains constant. By combining these two observations together we obtain:

$$\lim_{n \to \infty} \frac{w'(n)}{w'_\infty(n)} = 1, \qquad (78)$$

where

$$w'_\infty(n) = n^{N-1} \int_{\Omega_\infty} \delta(\mathbf{z} - \mathbf{z}^*) \frac{\det[2\pi\frac{1}{n}\mathbf{H}_{\mathbf{z}^*}^{-1}]^{1/2}}{\det[2\pi_{\mathbf{z}^*}]^{1/2}} e^{-g(\mathbf{z}^*)} e^{-nf(\mathbf{z}^*) - n\mathbf{R}(\mathbf{z},\mathbf{z}^*)}$$
$$= n^{N-1} \frac{\det[2\pi\frac{1}{n}\mathbf{H}^{-1}(\mathbf{z}^*)]^{1/2}}{\det[2\pi_{\mathbf{z}^*}]^{1/2}} e^{-g(\mathbf{z}^*)} e^{-nf(\mathbf{z}^*) - n\mathbf{R}(\mathbf{z}^*,\mathbf{z}^*)}. \qquad (79)$$

Note that by the definition, the residue vanishes at the expansion point: $\mathbf{R}(\mathbf{z}^*, \mathbf{z}^*) = 0$. Finally, by moving $2\pi$ and $\frac{1}{n}$ outside the determinants, one obtains

$$w'_\infty(n) = C_0 n^{-1/2} e^{-nC_2}, \qquad (80)$$

where $C_0 = \left(\frac{\det[\mathbf{H}_{\mathbf{z}^*}^{-1}]}{2\pi\det[\sigma_{\mathbf{z}^*}]}\right)^{1/2} e^{-g(\mathbf{z}^*)}$ and $C_2 = f(\mathbf{z}^*)$.

In what follows, we will consider the asymptote for the complete size distribution. The characteristic function for (49) is given by:

$$D_{i,j}(\boldsymbol{\omega}) = \delta_{i,j} + i\frac{\partial}{\partial\omega_j}\phi_i(\boldsymbol{\omega})\phi_i^{-1}(\boldsymbol{\omega}), \qquad (81)$$

so that the characteristic function of the full expression appearing under the sum in (47) is given by $\det[\mathbf{D}(\boldsymbol{\omega})]\phi(\boldsymbol{\omega})\prod_{l=1}^N \phi_l^{k_i}(\boldsymbol{\omega}) = \det[\mathbf{D}'(\boldsymbol{\omega})]$, where

$$D'_{i,j} := \delta_{i,j}\phi^{\frac{1}{N}}(\boldsymbol{\omega})\psi^N(\boldsymbol{\omega}, \mathbf{k}) + i\frac{\partial}{\partial\omega_j}\phi_i(\boldsymbol{\omega})\phi_i^{k_i-1}(\boldsymbol{\omega})\phi_i^{\frac{1}{N}}(\boldsymbol{\omega})\prod_{l=\{1,\ldots N\}\setminus i}\phi_l^{k_l}(\boldsymbol{\omega})$$
$$= \delta_{i,j}\phi^{\frac{1}{N}}(\boldsymbol{\omega})\psi^N(\boldsymbol{\omega}, \mathbf{k}) + i\frac{N}{k_i}\frac{\partial}{\partial\omega_j}\phi_i^{\frac{k_i}{N}}(\boldsymbol{\omega})\phi_i^{\frac{1}{N}}(\boldsymbol{\omega})\prod_{l=\{1,\ldots,N\}\setminus i}\phi_l^{\frac{k_l}{N}}(\boldsymbol{\omega}). \qquad (82)$$

The limiting functions for $\phi_i(\boldsymbol{\omega})$ are given in (61), so that one can write

$$\phi_{i,\infty}^{\frac{k_i}{N}}(\boldsymbol{\omega}) = e^{\frac{k_i}{N}i\boldsymbol{\mu}_i^\top \boldsymbol{\omega} - \frac{1}{2}\boldsymbol{\omega}^\top\frac{k_i}{N}\mathbf{T}_i\boldsymbol{\omega}}, \qquad (83)$$

which feature the following partial derivatives

$$i\frac{N}{k_i}\frac{\partial}{\partial\omega_j}\phi_{i,\infty}^{\frac{k_i}{N}}(\boldsymbol{\omega}) = -(\boldsymbol{\mu}_i^\top - i\boldsymbol{\omega}^\top\mathbf{T}_i)\mathbf{e}_j\phi_{i,\infty}^{\frac{k_i}{N}}(\boldsymbol{\omega}), \qquad (84)$$

By replacing $\phi_i(\boldsymbol{\omega})$ and $i\frac{N}{k_i}\frac{\partial}{\partial\omega_j}\phi_i(\boldsymbol{\omega})$ with their limiting functions, (83) and (84), we obtain a chain of transformations:

$$D'_{i,j} = \delta_{i,j}\phi(\boldsymbol{\omega})\psi_\infty^{\frac{1}{N}}(\boldsymbol{\omega}, \mathbf{k}) + i\frac{N}{k_i}\frac{\partial}{\partial\omega_j}\phi_{i,\infty}^{\frac{k_i}{N}}(\boldsymbol{\omega})\phi(\boldsymbol{\omega})\prod_{l=\{1,\ldots,N\}\setminus i}\phi_{l,\infty}^{\frac{k_l}{N}}(\boldsymbol{\omega})$$
$$= \delta_{i,j}\phi(\boldsymbol{\omega})\psi_\infty^{\frac{1}{N}}(\boldsymbol{\omega}, \mathbf{k}) - (\boldsymbol{\mu}_i^\top - ^\top \mathbf{T}_i)\mathbf{e}_j\phi_{i,\infty}^{\frac{k_i}{N}}(\boldsymbol{\omega})\phi(\boldsymbol{\omega})\prod_{l=\{1,\ldots,N\}\setminus i}\phi_{l,\infty}^{\frac{k_l}{N}}(\boldsymbol{\omega})$$
$$= \delta_{i,j}\phi(\boldsymbol{\omega})\psi_\infty^{\frac{1}{N}}(\boldsymbol{\omega}, \mathbf{k}) - (\boldsymbol{\mu}_i^\top - i\boldsymbol{\omega}^\top\mathbf{T}_i)\mathbf{e}_j\psi_\infty(\boldsymbol{\omega}, \mathbf{k})$$
$$= \phi^{\frac{1}{N}}(\boldsymbol{\omega})\psi_\infty^{\frac{1}{N}}(\boldsymbol{\omega}, \mathbf{k})\left[\delta_{i,j}(\boldsymbol{\mu}_i^\top - i\boldsymbol{\omega}^\top\mathbf{T}_i)\mathbf{e}_j\right]. \qquad (85)$$

Determinant $\det[D']$ can be now rewritten in the matrix form:

$$\det[\mathbf{D}'] = \psi_\infty(\boldsymbol{\omega}, \mathbf{k})\det(\mathbf{I} - \mathbf{M})\det[\mathbf{D}''], \quad D''_{i,j} = \delta_{i,j} + i\boldsymbol{\omega}^\top\mathbf{t}_{i,j}, \qquad (86)$$

where $\mathbf{t}_{i,j} = (\mathbf{I} - \mathbf{M})^{-1}\mathbf{T}_i\mathbf{e}_j$. Let us expand determinant $\det[\mathbf{D}']$ into the sum over $S_N$, the set of all permutations of $\{1, 2, \ldots, N\}$:

$$\det[\mathbf{D}'] = \psi_\infty(\boldsymbol{\omega}, \mathbf{k})\det(\mathbf{I} - \mathbf{M})\sum_{\sigma \in S_N}\text{sgn}(\sigma)\prod_{i=1}^N(\delta_{i,\sigma_i} + i\boldsymbol{\omega}^\top\mathbf{t}_{i,\sigma_i})$$
$$= \psi_\infty(\boldsymbol{\omega}, \mathbf{k})\det(\mathbf{I} - \mathbf{M})\left(\mathbf{c}_0 + i\boldsymbol{\omega}^\top\mathbf{c}_{1,1} + (i\boldsymbol{\omega}^\top\mathbf{c}_{2,1})(i\boldsymbol{\omega}^\top\mathbf{c}_{2,2}) + \ldots + \prod_{i=1}^N i\boldsymbol{\omega}^\top\mathbf{c}_{N,i}\right), \qquad (87)$$

where $\mathbf{c}_{i,j} \in \mathbb{R}^N$. Since the gradient of $\psi_\infty(\boldsymbol{\omega}, \mathbf{k})$ is given by

$$\nabla\psi_\infty(\boldsymbol{\omega}, \mathbf{k}) = [i(\mathbf{M}\mathbf{k} + \boldsymbol{\mu}_0) - \sigma(\mathbf{k})]\boldsymbol{\omega}\psi_\infty(\boldsymbol{\omega}, \mathbf{k}), \qquad (88)$$

one can express the $i\boldsymbol{\omega}\psi_\infty(\boldsymbol{\omega}, \mathbf{k})$ from the latter equation as:

$$i\boldsymbol{\omega}\psi_\infty(\boldsymbol{\omega}, \mathbf{k}) = -\sigma^{-1}(\mathbf{k})\mathbf{M}\mathbf{k}\psi_\infty(\boldsymbol{\omega}, \mathbf{k}) - \sigma^{-1}(\mathbf{k})\boldsymbol{\mu}_0\psi_\infty(\boldsymbol{\omega}, \mathbf{k}) - i\sigma^{-1}(\mathbf{k})\nabla\psi_\infty(\boldsymbol{\omega}, \mathbf{k}), \qquad (89)$$

which is the characteristic function for

$$\mathbf{x}_n(\mathbf{z}) = \frac{1}{n}\sigma_\mathbf{z}^{-1}(n(\mathbf{M}-\mathbf{I})\mathbf{A}^{-1}\mathbf{z} + \boldsymbol{\mu}_0) = -\sigma_\mathbf{z}^{-1}(\mathbf{M}-\mathbf{I})\mathbf{A}^{-1}\mathbf{z} + \frac{1}{n}\sigma_\mathbf{z}^{-1}\boldsymbol{\mu}_0. \quad (90)$$

Since series (87) is the sum of powers of $i\boldsymbol{\omega}\psi_\infty(\boldsymbol{\omega}, \mathbf{k})$, this series is the characteristic function for an identical expression in which $i\boldsymbol{\omega}\psi_\infty(\boldsymbol{\omega}, \mathbf{k})$ is substituted by (90):

$$d(n, \mathbf{z}) = \det(\mathbf{I}-\mathbf{M})\left(\mathbf{c}_0 + \mathbf{x}_n^\top(\mathbf{z})\mathbf{c}_{1,1} + (\mathbf{x}_n^\top(\mathbf{z})\mathbf{c}_{2,1})(\mathbf{x}_n^\top(\mathbf{z})\mathbf{c}_{2,2}) + \dots + \prod_{i=1}^N \mathbf{x}_n^\mathrm{T}(\mathbf{z})\mathbf{c}_{N,i}\right)F_n(\mathbf{z}). \quad (91)$$

The latter expression is the product of $F_n(\mathbf{z})$ and a polynomial in the indeterminate $y = \frac{1}{n}$:

$$d(n, \mathbf{z}) = (a_0(\mathbf{z}) + a_1(\mathbf{z})y + a_2(\mathbf{z})y^2 + \dots + a_N(\mathbf{z})y^N)F_n(\mathbf{z}). \quad (92)$$

Instead of this polynomial, it is convenient to consider its collapsed form that we obtain by observing that the series from Eqs. (87) and (91) coincide under the substitution $i\boldsymbol{\omega} \to \mathbf{x}_n^\top(\mathbf{z})$, and therefore performing the same substitution in Eq. (86) leads to: $d(n, \mathbf{z}^*) = p(n)F_n(\mathbf{z}^*)$, where $p(n)$ is independent of $\mathbf{z}^*$:

$$p(n) = \det(\mathbf{M}-\mathbf{I})\det\left(\mathbf{I} + \mathbf{Q} + \frac{1}{n}\mathbf{R}\right), \quad (93)$$

square matrices $Q_{i,j} = \delta_{i,j} - [1, \mathbf{z}^*]^\top \mathbf{K}^\top \sigma_{\mathbf{z}^*}^{-1}\mathrm{adj}(\mathbf{M}-\mathbf{I})\mathbf{T}_i\,\mathbf{e}_j$, $R_{i,j} = \boldsymbol{\mu}_0^\top \sigma_{\mathbf{z}^*}^{-1}\mathrm{adj}(\mathbf{M}-\mathbf{I})\mathbf{T}_i\,\mathbf{e}_j$, are of size $N \times N$, and $(\mathbf{e}_i)_j = \delta_{i,j}$ are the standard basis vectors. One can derive the coefficients of the expansion (92) by applying differential operator $\frac{1}{k!}\frac{\partial^k}{\partial y^k}|_{y=0}$ to Eq. (93). For $k=0,1$ this procedure yields, $a_0 = \det(\mathbf{M}-\mathbf{I})\det[\mathbf{Q}] = 0$, and $a_1 = \det(\mathbf{M}-\mathbf{I})\mathrm{tr}(\mathrm{adj}(\mathbf{Q})\mathbf{R}) > 0$. By taking into the account the asymptotical behaviour of $F_n(\mathbf{z})$ as given by (80), the complete expression for the asymptote of (47) reads:

$$w_\infty(n) = p(n)C_0 n^{-1/2}e^{-C_2 n} = [a_1 n^{-1} + O(n^{-2})]C_0 n^{-1/2}e^{-C_2 n} \approx C_1 n^{-3/2}e^{-C_2 n}, \quad (94)$$

where

$$C_1 = C_0 a_1 = \det(\mathbf{M}-\mathbf{I})\mathrm{tr}(\mathrm{adj}(\mathbf{Q})\mathbf{R})\left(\frac{\det[\mathbf{H}_{\mathbf{z}^*}^{-1}]}{2\pi\det[\sigma_{\mathbf{z}^*}]}\right)^{1/2}e^{-[1,\mathbf{z}^{*\top}]\mathbf{K}^\top \sigma_{\mathbf{z}^*}^{-1}\boldsymbol{\mu}_0}. \quad (95)$$

and

$$C_2 = \frac{1}{2}[1, \mathbf{z}^{*\top}]\mathbf{K}^\top \sigma_{\mathbf{z}^*}^{-1}\mathbf{K}\begin{bmatrix}1\\\mathbf{z}^*\end{bmatrix}. \quad (96)$$

**Necessary and sufficient criteria for scale-free behaviour.** In this section of Methods, we prove that $C_2 = 0$ if and only if

$$\mathbf{v} \in \ker(\mathbf{M}-\mathbf{I}), \quad \frac{\mathbf{v}}{|\mathbf{v}|} \geq 0. \quad (97)$$

According to the definition (68), $\sigma_\mathbf{z}$ is a linear combination of covariance matrices, and therefore $\sigma_\mathbf{z}^{-1}$ is a positive definite matrix that features the Cholesky factorisation:

$$\sigma_\mathbf{z}^{-1} = \mathbf{L}^\top \mathbf{L}, \quad \det \mathbf{L} \neq 0. \quad (98)$$

Suppose $C_2 = 0$. Then, according to the definition of $C_2$, $f(\mathbf{z}^*) = 0$ and therefore:

$$[1, \mathbf{z}^{*\top}]\mathbf{K}^\top \sigma_{\mathbf{z}^*}^{-1}\mathbf{K}\begin{bmatrix}1\\\mathbf{z}^*\end{bmatrix} = [1, \mathbf{z}^{*\top}]\mathbf{K}^\top \mathbf{L}^\top \mathbf{L}\mathbf{K}\begin{bmatrix}1\\\mathbf{z}^*\end{bmatrix} = \left(\mathbf{L}\mathbf{K}\begin{bmatrix}1\\\mathbf{z}^*\end{bmatrix}\right)^\top \mathbf{L}\mathbf{K}\begin{bmatrix}1\\\mathbf{z}^*\end{bmatrix} = 0. \quad (99)$$

Recall that $\mathbf{K} = (\mathbf{M}-\mathbf{I})\mathbf{A}^{-1}$ and since $\det \mathbf{L} \neq 0$ we have:

$$\mathbf{K}\begin{bmatrix}1\\\mathbf{z}^*\end{bmatrix} = (\mathbf{M}-\mathbf{I})\mathbf{A}^{-1}\begin{bmatrix}1\\\mathbf{z}^*\end{bmatrix} = (\mathbf{M}-\mathbf{I})\mathbf{v} = 0, \quad \mathbf{v} = \mathbf{A}^{-1}\begin{bmatrix}1\\\mathbf{z}^*\end{bmatrix}. \quad (100)$$

So that $\mathbf{v} \in \ker(\mathbf{M}-\mathbf{I})$.

According to the definition (65), $|\mathbf{v}| = 1 - \sum_{i=2}^N z_i^* + \sum_{i=2}^N z_i^* = 1$. We will now demonstrate that $\mathbf{v} > 0$. Assume that $\mathbf{z}^* \notin \Omega_\infty$, then for any $\mathbf{z} \in \Omega_\infty$, $f(\mathbf{z}) > 0$, and

therefore for arbitrary $\alpha > 0$ and non-singular function $\gamma(\mathbf{z})$ the following product vanish in the limit of large $n$,

$$\lim_{n \to \infty} n^\alpha \int_{\Omega_n} \gamma(\mathbf{z})e^{-\frac{1}{2}nf(\mathbf{z})}\mathrm{d}\mathbf{z} \to 0, \quad (101)$$

which contradicts our assumption that the asymptote is scale-free. One therefore concludes that $\mathbf{z}^* \in \Omega_\infty$, so that the set of inequalities (76) holds for $\mathbf{z}^*$. This inequalities imply: $v_1 = 1 - |\mathbf{z}^*| > 0$ and $v_i = z_{i-1} > 0$, for $i = 2, 3, \dots, N$, and therefore $\frac{\mathbf{v}}{|\mathbf{v}|} = \mathbf{v} > 0$. This completes the proof of the forward implication.

Consider the reverse implication: suppose $\mathbf{v} \in \ker(\mathbf{M}-\mathbf{I})$, and $\frac{\mathbf{v}}{|\mathbf{v}|} > 0$. Without loss of generality, assume $|\mathbf{v}| = 1$. We will demonstrate that vector $\mathbf{z}^* = (v_2, v_3, \dots, v_N)^\mathrm{T}$ gives minimum to $f(\mathbf{z})$, and $f(\mathbf{z}^*) = 0$. On the one hand, we have a chain of transformations that is reverse to (99):

$$0 = (\mathbf{M}-\mathbf{I})\mathbf{v} = \mathbf{K}\begin{bmatrix}1\\\mathbf{z}^*\end{bmatrix} = \mathbf{L}\mathbf{K}\begin{bmatrix}1\\\mathbf{z}^*\end{bmatrix} = \left(\mathbf{L}\mathbf{K}\begin{bmatrix}1\\\mathbf{z}^*\end{bmatrix}\right)^\top \mathbf{L}\mathbf{K}\begin{bmatrix}1\\\mathbf{z}^*\end{bmatrix}$$
$$= [1, \mathbf{z}^{*\top}]\mathbf{K}^\top \mathbf{L}^\top \mathbf{L}\mathbf{K}\begin{bmatrix}1\\\mathbf{z}^*\end{bmatrix} = [1, \mathbf{z}^{*\top}]\mathbf{K}^\top \sigma_{\mathbf{z}^*}^{-1}\mathbf{K}\begin{bmatrix}1\\\mathbf{z}^*\end{bmatrix} = f(\mathbf{z}^*).$$

On the other hand, since $f(\mathbf{z}) \geq 0$ for $\mathbf{z} \geq 0$, $\mathbf{z}^*$ must also be a minimum of $f(\mathbf{z})$. Furthermore, this minimum belongs to $\Omega_\infty$. Indeed, $\mathbf{z}_i^* = v_{i+1} > 0$ for $i = 1, \dots, N-1$, and $|\mathbf{z}^*| = \sum_{i=2}^N v_i = |\mathbf{v}| - v_1 < 1$. The latter inequalities imply that $\mathbf{z}^* \in \Omega_\infty$, which guarantees validity of asymptote (94), and since $C_2 = f(\mathbf{z}^*) = 0$, this asymptote is a scale-free one.

**Derivations of the critical percolation probability.** From the perspective of a randomly chosen node, each adjacent edge has equal and independent chances to be removed, so that the actual degree distribution after percolation can be expressed by multiplying $u(\mathbf{k})$ with the binomial distribution:

$$u'(\mathbf{k}') = \sum_{\mathbf{k} \geq 0}\prod_{i=1}^N \binom{k_i}{k_i'}p^{k_i'}(1-p)^{k_i-k_i'}u(\mathbf{k}). \quad (102)$$

The expectations of $u'(\mathbf{k}')$ and those of $u(\mathbf{k})$ are related:

$$\mathbb{E}[k_i'] = p\mathbb{E}[k_i], \quad (103)$$

$$\mathbb{E}[k_i', k_j'] = p^2\mathbb{E}[k_i, k_j] + (p - p^2)\mathbb{E}[k_i]\delta_{i,j}, \quad (104)$$

$$\begin{aligned}\mathbb{E}[k_i', k_j', k_l'] &= p_i p_j p_l \mathbb{E}[k_i, k_k, k_l] + p_i p_i(1-p_i)\mathbb{E}[k_i, k_l]\delta_{i,j} \\ &\quad + p_i p_j(1-p_i)\mathbb{E}[k_i, k_j]\delta_{i,l} + p_j p_i(1-p_j)\mathbb{E}[k_j, k_i]\delta_{j,l} \\ &\quad + p_i(1 - 3p_i + 2p_i^2)\mathbb{E}\delta_{i,j}\delta_{i,l},\end{aligned} \quad (105)$$

and by plugging these substitutions into (43), (45) and (46) one obtains:

$$\boldsymbol{\mu}' = p\boldsymbol{\mu}, \quad (106)$$

$$\mathbf{M}' = p\mathbf{M}, \quad (107)$$

$$(\mathbf{T}_i')_{j,l} = p^2(\mathbf{T}_i)_{j,l} + p(1 - p)M_{j,i}\delta_{j,l}. \quad (108)$$

These can be now used to compute the asymptotic properties of the percolated network. For instance, by plugging $\mathbf{M}'$ into the criticality criterion (97), one obtains a $p$-dependant criterion that reads: the edge-coloured network features critical percolation at $p = p_c \in (0, 1]$ if there is vector $\mathbf{v}$, for which

$$\mathbf{v} \in \ker[p_c\mathbf{M} - \mathbf{I}], \quad \frac{\mathbf{v}}{|\mathbf{v}|} \geq 0. \quad (109)$$

**Derivations for colour-dependant percolation.** In colour-dependent percolation, the percolation probability depends on the colour of an edge. In this case, we consider a vector $\mathbf{p} = (p_1, p_2, \dots, p_N)^\top$, where $p_i$ is the probability that an edge of colour $i$ is not removed. Colour-dependant percolation affects the degree

distribution, so that after the percolation process the degree distribution becomes:

$$u'(\mathbf{k}') = \sum_{\mathbf{k} \geq 0} \prod_{i=1}^{N} \binom{k_i}{k_i'} p_i^{k_i'} (1 - p_i)^{k_i - k_i'} u(\mathbf{k}). \tag{110}$$

Computing the expectations for the latter distribution gives:

$$\mathbb{E}[k_i'] = p_i \mathbb{E}[k_i], \tag{111}$$

$$\mathbb{E}[k_i', k_j'] = p_i p_j \mathbb{E}[k_i, k_j] + (p_i - p_i^2) \mathbb{E}[k_i] \delta_{i,j}, \tag{112}$$

$$\begin{aligned}
\mathbb{E}[k_i', k_j', k_l'] &= p_i p_j p_l \mathbb{E}[k_i, k_k, k_l] + p_l p_i (1 - p_i) \mathbb{E}[k_i, k_l] \delta_{ij} \\
&+ p_i p_j (1 - p_i) \mathbb{E}[k_i, k_j] \delta_{i,l} + p_j p_i (1 - p_j) \mathbb{E}[k_j, k_i] \delta_{i,l} \\
&+ p_i (1 - 3p_i + 2p_i^2) \mathbb{E}[k_i] \delta_{i,j} \delta_{i,l},
\end{aligned} \tag{113}$$

and plugging these into Eqs. (45)–(46) allows us to express $\boldsymbol{\mu}$, $\boldsymbol{M}\, \boldsymbol{T}_i$ as functions of $\mathbf{p}$:

$$\boldsymbol{\mu}' = \text{diag}\{\mathbf{p}\}\boldsymbol{\mu}, \tag{114}$$

$$\boldsymbol{M}' = \text{diag}\{\mathbf{p}\}\boldsymbol{M}, \tag{115}$$

$$\boldsymbol{T}_t' = \text{diag}\{\mathbf{p}\}\boldsymbol{T}_i \text{diag}\{\mathbf{p}\} + \text{diag}\{\mathbf{p}\}\text{diag}\{1 - \mathbf{p}\}\text{diag}\{M_{1,j}, \dots, M_{N,j}\}. \tag{116}$$

By plugging $\boldsymbol{M}'$ into Eq. (97) one obtains the criterion for criticality: edge-coloured network features a critical behaviour at percolation probability vector $0 < \mathbf{p} < 1$ if and only if:

$$\mathbf{z} \in \ker[\text{diag}\{\mathbf{p}\}\boldsymbol{M} - \boldsymbol{I}], \frac{\mathbf{z}}{|\mathbf{z}|} \geq 0. \tag{117}$$

In order to recover the manifold containing all critical points $\mathbf{p}$ numerically, one may follow this practical procedure:

1. for all the points from a discretised unit hypercube test if $\det(\text{diag}\{\mathbf{p}\}\boldsymbol{M} - \boldsymbol{I}) = 0$;
2. for those points that pass the first test, find the eigenpair $(\mathbf{v}, 0)$ of $\text{diag}\{\mathbf{p}\}\boldsymbol{M} - \boldsymbol{I}$ and check whether $\frac{\mathbf{v}}{|\mathbf{v}|} > 0$.

**Derivation of colour fractions in finite components**. For the same reason as in the conventional configuration model, every connected component of finite size $n$ has a tree-like structure, and therefore this component contains $n - 1$ edges. The generalised model operates with $N$ types of edges and it is interesting to see how these $n - 1$ edges are partitioned among $N$ edge types. It turns out that $\mathbf{z}^*$, as defined in (69), plays an essential role in defining this partition. Let $v_i$, $i = 1, \dots, N$ denotes the number of edges of $i^{\text{th}}$ type in a component of size $n$. Since the total number of edges is $|\mathbf{v}| = n - 1$, one writes $v_1 = 1 - \sum_{i=2}^{N} v_i$, and the probabilities of configurations for the rest of edge types $v_2, v_3, \dots v_N$ follow form $F_n(\mathbf{z}^*)$ as defined in Eq. (73). In fact, since we are interested in the conditional probability given component size is $n$, it is enough to consider the first fraction appearing in (73). So that the following law of large numbers holds:

$$\begin{aligned}
&\mathbb{P}[\lfloor nz_1 \rfloor \leq v_2 \leq \lceil nz_1 \rceil \wedge, \dots, \wedge \lfloor nz_{N-1} \rfloor \leq v_N \leq \lceil nz_{N-1} \rceil | \text{component size} = n] \\
&= \int_{\lfloor nz \rfloor}^{\lceil nz \rceil} \frac{e^{-\frac{1}{2}(z - z^*)^\top \left[\frac{1}{n}\mathbf{H}_{z^*}^{-1}\right]^{-1}(z - z^*)}}{\det\left[2\pi \frac{1}{n}\mathbf{H}_{z^*}^{-1}\right]^{1/2}} \approx \frac{e^{-\frac{1}{2}(z - z^*)^\top \left[\frac{1}{n}\mathbf{H}_{z^*}^{-1}\right]^{-1}(z - z^*)}}{\det\left[2\pi \frac{1}{n}\mathbf{H}_{z^*}^{-1}\right]^{1/2}}.
\end{aligned} \tag{118}$$

Evidently, the multivariate stochastic variable $\left(\frac{v_2}{n-1}, \frac{v_3}{n-1}, \dots, \frac{v_N}{n-1}\right)$ is normally distributed with mean $\mathbf{z}^*$ and variance $\frac{1}{n}\mathbf{H}_{z^*}$, and the whole vector $\frac{\mathbf{v}}{n-1}$ is normally distributed, with probability density $\mathcal{N}\left(\mathbf{f}, \mathbf{m}, \frac{1}{n}\boldsymbol{\Sigma}\right)$ having mean vector

$$\mathbf{m} = (1 - |\mathbf{z}^*|, z_1^*, z_2^*, \dots, z_{N-1}^*) \tag{119}$$

and covariance matrix

$$\boldsymbol{\Sigma} = \frac{1}{n} \begin{bmatrix} a & \mathbf{b}^\top \\ \mathbf{b} & \mathbf{H}_{z^*} \end{bmatrix}, \tag{120}$$

where $a = \sum_{i,j=1}^{N-1} (\mathbf{H}_{z^*})_{i,j}$, and $\mathbf{b}$ is a column vector of length $N - 1$,

$$b_i = -\sum_{j=1}^{N-1} (\mathbf{H}_{z^*})_{i,j}.$$

**Derivation of the size of the giant component**. Let random variable $n$ is the size of the component containing a randomly chosen node. Naturally, we assume that nodes are sampled uniformly at random. The generating function for $n$, $W(x)$ satisfies the system of $N$ equations:

$$\begin{cases}
W(x) &= xU[W_1(x), \dots, W_N(x)], \\
W_1(x) &= xU_1[W_1(x), \dots, W_N(x)], \\
\dots \\
W_N(x) &= xU_N[W_1(x), \dots, W_N(x)].
\end{cases} \tag{121}$$

These equations constitute a generalisation of the corresponding system introduced for uncoloured networks[19]. In Eq. (121), $W_i(x)$ generate probabilities that a randomly selected node is connected to a component of size $m$ on either of its sides. Since the network may contain infinitely large components, $n$ and $m$ are improper random variables. We formalise this facts by writing:

$$g_{\text{node}} := 1 - W(1) = \mathbb{P}[n = \infty] \geq 0, \tag{122}$$

$$s_i := W_i(1) = 1 - \mathbb{P}[m = \infty] \leq 1. \tag{123}$$

Plugging these definitions into (121) yields the following equations:

$$g_{\text{node}} = 1 - \mathbb{E}[\mathbf{s}^\mathbf{k}], \mathbf{s} = (s_1, \dots, s_N)^\top, \tag{124}$$

and

$$s_i = \frac{\mathbb{E}[k_i \mathbf{s}^{\mathbf{k} - \mathbf{e}_i}]}{\mathbb{E}[k_i]}, i = 1, \dots, N. \tag{125}$$

Here, $\mathbf{e}_i$ are the standard basis vectors, and vector power is evaluated element-wisely: $\mathbf{s}^\mathbf{k} = \prod_{i=1}^{N} s_i^{k_i}$. The quantities $g_{\text{node}}$ and $\mathbf{s}$ have straightforward interpretations: $g_{\text{node}}$, or the node size of the ginat component, is the probability that a randomly sampled node belongs to the giant component; $s_i$ are the probabilities that a randomly sampled $i$-coloured edge is not connected to a giant component on at least one side. So that the edge size of the giant component is written as the probability that a randomly chosen edge of colour $i$ belongs to the giant component, $g_i = 1 - s_i^2$, $i = 1, \dots, N$. By weighting these numbers with the total fractions of coloured edges $c_i = \mathbb{E}[k_i] / \sum_{i=1}^{N} \mathbb{E}[k_i]$, one obtains the vector of colour fractions in the giant component,

$$\mathbf{v}^* = (v_1^*, v_2^*, \dots, v_N^*), v_i^* = \frac{g_i c_i}{\mathbf{g}^\top \mathbf{c}}, \tag{126}$$

which is the giant-component analog of (118). Since the giant component is infinite by definition, $\mathbf{v}^*$ is not a random variable as is the case with colour fractions in finite components (118).

**Derivation of the expected size of finite components**. We will now give a quantitive estimate for the size of typical finite connected component. Formally speaking, we wish to extract some properties of the size of uniformly at randomly chosen component. Let us denote this size as random variable $n'$. It is important to note the subtle difference between $n'$ and $n$—they differ in the method of sampling. For the former, we first chose a node, and then take the size of the component the node belongs to, for the latter—we chose the component itself.

Given the tools at hand, it is not easy to derive the expression for average component size $\mathbb{E}[n']$. With little efforts, however, we can derive one for the ratio:

$$w_{\text{avg}} := \frac{\mathbb{E}[n'^2]}{\mathbb{E}[n']} = \frac{\mathbb{E}[n]}{1 - g_{\text{node}}} = \frac{\frac{d}{dx}W(x)|_{x=1}}{1 - g_{\text{node}}}. \tag{127}$$

In polymer literature, this ratio is commonly referred to as the weight-average size[16] and we thus conveniently borrow this terminology. The weight-average size of finite connected components $w_{\text{avg}}$ is found by first expressing the derivatives of $W_i(x)$ and $W(x)$ from Eq. (125).

Let $\mathbf{y} = (y_1, y_2, \ldots, y_N)^\top$, $y_i = \frac{d}{dx} W_i(x)|_{x=1}$, then

$$y_i = \left( W_1'(x) \frac{\partial}{\partial s_1} U_i[W_1(x), \ldots, W_N(x)] + \ldots + W_N'(x) \frac{\partial}{\partial s_N} U_i[W_1(x), \ldots, W_N(x)] \right)|_{x=1}$$
$$+ U_i[W_1(1), \ldots, W_N(1)], \tag{128}$$

or alternatively in the matrix form: $\mathbf{y} = [\mathbf{I} - \mathbf{X}(\mathbf{s})]^{-1}\mathbf{s}$, where

$$X_{i,j}(\mathbf{s}) = \frac{\partial}{\partial s_j} U_i[W_1(x), \ldots, W_N(x)]|_{x=1} = \frac{\mathbb{E}[(k_i k_j - \delta_{i,j} k_i) \mathbf{s}^{\mathbf{k} - \mathbf{e}_i - \mathbf{e}_j}]}{\mathbb{E}[k_i]}, \quad i, j = 1, \ldots, N. \tag{129}$$

In a similar fashion one obtains the expression for $W'(1)$:

$$\frac{d}{dx} W(x)|_{x=1} = \left( W_1'(x) \frac{\partial}{\partial s_1} U[W_1(x), \ldots, W_N(x)] + \ldots + W_N'(x) \frac{\partial}{\partial s_N} U[W_1(x), \ldots, W_N(x)] \right)|_{x=1}$$
$$+ U[W_1(1), \ldots, W_N(1)] = \mathbf{s}^\top \mathbf{D} \mathbf{y} + 1 - g_{\text{node}}, \mathbf{D} = \text{diag}\{\mathbb{E}[k_1], \ldots, \mathbb{E}[k_N]\}. \tag{130}$$

So that

$$w_{\text{avg}} = \frac{\frac{d}{dx} W(x)|_{x=1}}{1 - g_{\text{node}}} = \frac{\mathbf{s}^\top \mathbf{D}[\mathbf{I} - \mathbf{X}(\mathbf{s})]^{-1}\mathbf{s}}{1 - g_{\text{node}}} + 1. \tag{131}$$

See also Ref. [16] for the derivation of the wight-average size of connected components in the case of unicoloured networks.

Finally, we show how expressions (132) and (131) change as a result of simple bond percolation. In this case, the degree distribution becomes dependant on percolation probability $p$ as defined in Eq. (102), which induces the following transformation of the giant component size:

$$g_{\text{node}}(p) = 1 - \mathbb{E}[(p(\mathbf{s}_p - 1) + 1)^{\mathbf{k}}], \tag{132}$$

where $(s_p)_i = \frac{\mathbb{E}[k_i(p(\mathbf{s}_p - 1) + 1)^{\mathbf{k} - \mathbf{e}_i}]}{\mathbb{E}[k_i]}, i = 1, \ldots, N$. In a similar fashion, $w_{\text{avg}}$ also becomes a function of $p$:

$$w_{\text{avg}}(p) = \frac{\mathbf{s}_p^\top \mathbf{D}[p^{-1}\mathbf{I} - X(p(\mathbf{s}_p - 1) + 1)]^{-1}\mathbf{s}_p}{1 - g_{\text{node}}(p)} + 1. \tag{133}$$

**The first singularity of the weight-average component size**. Let $p_c$ be the earliest critical point, that is to say the smallest value that solves Eq. (109). When $p < p_c$, the giant component does not exist, and we have $\mathbf{s}_p = \mathbf{1}$, $g_{\text{node}}(p) = 0$ and consequently $X(p(\mathbf{s}_p - 1) + 1) = X(\mathbf{1}) = \mathbf{M}'$. By plugging these expressions into Eq. (133) gives:

$$w_{\text{avg}}(p) = \mathbf{1}^\top \mathbf{D}[p^{-1}\mathbf{I} - \mathbf{M}']^{-1}\mathbf{1} + 1, \quad p < p_c. \tag{134}$$

Let $\mathbf{M} = \mathbf{P} \text{diag}\{\lambda_i\} \mathbf{P}^{-1}$ be the eigenvalue decomposition of $\mathbf{M}$, then

$$[p^{-1}\mathbf{I} - \mathbf{M}']^{-1} = \mathbf{P}^{-1} \text{diag}\left\{ \frac{1}{p^{-1} - \lambda_i} \right\} \mathbf{P} = \mathbf{P}^{-1} \text{diag}\left\{ \frac{p/\lambda_i}{\lambda_i^{-1} - p} \right\} \mathbf{P}, \tag{135}$$

and therefore $w_{\text{avg}}(p)$ can be represented as a linear combination with finite coefficients:

$$w_{\text{avg}}(p) = \sum_{i=1}^N \frac{c_i}{\lambda_i^{-1} - p}, \quad c_i = \sum_j (\mathbf{D}\mathbf{P}^{-1})_{j,i} \sum_k (\mathbf{P})_{i,k} < \infty. \tag{136}$$

Since there is such $i$ that $p_c = \lambda_i^{-1}$, it becomes clear that $w_{\text{avg}}(p)$ diverges at $p = p_c$. Moreover, the singularity is characterised by $\lim_{p \to p_c} \frac{w_{\text{avg}}(p)}{p_c - p} = \mathcal{O}(1)$, which is identical to the case of unicoloured networks[16].

## Data availability
The source code reproducing the examples is available at the GitHub repository: https://github.com/ikryven/ColorPercolation. The datasets generated during the current study are available from the author on reasonable request.

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

## Acknowledgements

This work was part of the research programme VENI with Project no. 639.071.511, which was financed by the Netherlands Organisation for Scientific Research (NWO).

## Author contributions

Conceived, implemented and written by I. Kryven.

## Additional information

**Competing interests:** The authors declare no competing interests.

