## [Peer Review File · Nature Communications]

Reviewers' comments:

Reviewer #1 (Remarks to the Author):

In this manuscript, Ivan Kryven investigates a generic analytic theory to describe how the percolation properties are affected in regular and colored networks. First he shortly introduces a reader to his theory (which heavily relies on his previous papers: "Ivan Kryven. Finite connected components in infinite directed and multiplex networks with arbitrary degree distributions. Physical Review E, 96(5):052304, 2017." and "Ivan Kryven. General expression for the component size distribution in infinite configuration networks. Physical Review E, 95(5):052303, 2017."). After that he lays out the framework for studying criticality and describes all the important quantities. He presents a number of interesting features found in model examples that he laid out in this paper. Most interesting are the manifolds containing critical configurations, and switching behavior of colors in finite components.

This manuscript presents very interesting results for the percolation in complex networks. The theory which the author have developed in this and previous two papers has a potential to become a go to technique for many scientists working in the field of complex networks. I am grateful that I was chosen to review this manuscript because I had to read and study the papers which I did not notice and which I find important and very useful. Having said that, I can not recommend publication of the manuscript in this form and in this stage.

First problem is the readability of the whole manuscript which could warrant the whole rewriting of the methods section and adding some additional supplementary information.

Second problem is that the major breakthrough is in fact the development of the theory which was already published in two aforementioned papers. The result of color switching is interesting, but is also a property of very simple model the author proposed. The fact that his theory can be used to understand model well does is nice, but the finding is property of the model. The text regarding this finding is also unclear.

Q1: Color switching is presented in simulations only for components of size 50. It is not really clear what happens in the giant component and what happens in finite components. I would assume that the whole fig 3 is for finite components, as u number of links of type 3 is always greater than the number of links of type one (if we assume that the nodes of type 1 and type 3 have the same probability). So in Giant component for $p=1$ I would assume that the fraction of type 4 dominates,

which is just the opposite to the figure (in which I would assume to have finite components dominated with type 1.

The presentation of critical manifolds is well written and understandable. This part of the manuscript is the one which would most influenced the field and the author should perhaps concentrate more effort on it. My problem with this part of manuscript is a certain "abuse" of language as author defines for small C_2 coefficients a "critical window" (notably for $C_2 = 0.02$). While results are interesting I strongly urge author to change the language because what is a small value for C_2 is just arbitrary in this text. Unless there is a clear mathematical criterion introduced, author should avoid this language, especially given the fact that the whole text is generally very precise and stringent from mathematical point of view.

Q3: How it would relate to other percolation properties of complex networks. For example can this theory be related to the scaling laws as found in: "Dong, Gaogao, et al. "Resilience of networks with community structure behaves as if under an external field." Proceedings of the National Academy of Sciences (2018): 201801588."

Furthermore there were previous reports of critical manifolds (take for example:

"Shao, Jia, et al. "Cascade of failures in coupled network systems with multiple support-dependence relations." Physical Review E 83.3 (2011): 036116." - can the author relate his theory to their findings?

There were also other approaches to model percolation of colored networks ("Krause, Sebastian M., et al."Color-avoiding percolation." Physical Review E 96.2 (2017): 022313.") does the approach presented here works in the case of this type of percolation?

Additional Questions:

Q2: There is a difference between EQ4 and Eq S7 in the supplement and I assume this are the same equations. Either D is an auxiliary function or is a matrix defined in S9 on which convolution determinant has to be computed.

Q3: I do not really understand the equation 16, does it mean that $u(1,2,3) = \text{Poiss}(20,3)$? If so it would be nice to explain to the reader why you choose this particular unusual distribution. Is it just for ease of computations or to show that theory works also for correlated distributions?

Q4: It would be nice if the author would provide a bit more of intuitive explanation to eq: 6 (it follows from analysis given in supplement, but one would like to get a feeling why the M matrix is so important for criticality). It is also possible that more intuitive explanation is not possible and in that case I would urge author to disregard this question.

Comments:

C1: Author likes to use an uncommon notation which makes his text harder to follow. Excess degree distribution is biased degree distribution etc. This can in principle stay in the paper, but I would strongly suggest to the author to change his nomenclature to a more common to its expected audience (for example to use any of notations and nomenclature from any of the references 3-7).

C2: Equation 6, 7 and 8 are very important and a feature of theory that SHOULD be stressed even more by the author.

Reviewer #2 (Remarks to the Author):

In this manuscript the author introduces interesting analytic calculations (and simulations supporting them) regarding connectivity of edge-colored multigraphs. The author reports that these calculations are building on the theory he published last year on Physical Review E. After deriving the general results, two seemingly arbitrarily chosen toy examples are analysed in great detail. The reported main results of this analysis include the finding multiple distinct percolation thresholds (exemplified by a case with 3 distinct percolation threshold when there are essentially 3 distinct networks), a particular pattern of change in the edge-color composition of non-giant components, and an illustration that the edge-occupation probabilities can be selected in a way that the system stays at a critical state when these probabilities are varied (i.e., instead of a critical point there is critical range).

The manuscript seems to stand on a strong theoretical basis, but the application of this advance in the theory remains speculative: there are several statements where this theory could be used, but the examples presented in the manuscript do not illustrate examples of these speculative problems. If the theory can be used to solve outstanding problems in the literature or change the way some systems are analysed, why not show few examples of this? Further, comparison to previous theory is almost missing (except for a paragraph in the introduction and few reference later). That is, this

manuscript is promising, but it is difficult to recommend publishing it Nature Communications if the reported impact of the results remain on such uncertain basis.

Further major comments:

A. Connection to the previous literature could be stronger. First of all, it would be good to mention that these types of networks have been studied earlier under the name of multiplex networks. In fact, the concept "coloured networks" can be slightly misleading as it is often used also to denote networks where the nodes are coloured. Further, it would be good to mention the work on mutual percolations/cascading failures this related (but different) percolation process has been under active study in recent years.

B. The connection of the theoretical results to previous very similar results are not explored properly. For example, Ref 16 defines a similar colored excess degree matrix to M and solves a very similar eigenvalue problem on it (see Eqs. 5 and 6 in ref 16, where the matrix B similar to M and an eigenvalue problem is defined). It is mentioned in subsection b that the results generalize the ones of equation 7 in ref 16, but the differences between the theories and their predictions should be made very clear to the reader.

C. Many real networks with multiple types of links contain large amounts of "edge overlap", where the same pairs of nodes are connected with multiple types of links. It seems that the formalism in this manuscript does not easily allow including this type of correlations in the model. This probably seriously limits the applicability of the results to many real-world systems, but could probably be introduced relatively easily to the equations(?). In any case this should be discussed in the manuscript.

D. The manuscript would be made stronger if the main results could be made more clear earlier. The sections A-F are rather dense and difficult to understand without following the references to the supplementary and methods, and things start to clear out only when the toy examples are shown in section F. Further, the results on the hierarchies of connectivities seem rather trivial as adding more links will always increase the connectivity, and the relevance and applicability of these results remain a mystery to me.

Minor comments (mostly typos):

1. Page 6, "To this end,": This is at the start of a paragraph, so it is not clear to what end.
2. Page 6, "any colours form index set": should it be "from"?
3. Page 7, notation of eq 12?
4. Page 10, Fig 2 caption: "vales" -> "values"
5. Page 12, Fig 3 caption: "faction" -> "fraction"

Reviewer #1

1.

This manuscript presents very interesting results for the percolation in complex networks. The theory which the author have developed in this and previous two papers has a potential to become a go to technique for many scientists working in the field of complex networks. I am grateful that I was chosen to review this manuscript because I had to read and study the papers which I did not notice and which I find important and very useful. Having said that, I can not recommend publication of the manuscript in this form and in this stage.

First problem is the readability of the whole manuscript which could warrant the whole rewriting of the methods section and adding some additional supplementary information.

The entire manuscript was considerably reworked to improve the flow of the arguments. The main results are moved closer to the beginning of Results and highlighted with additional explanatory text (pp. 4-5). Figure 1 and Fig. 2 were added to schematically introduce the setup of the study. Methods were merged with the Supplementary Information and a few additional Supplementary Sections were added (as explained in the replies below).

2.

Second problem is that the major breakthrough is in fact the development of the theory which was already published in two aforementioned papers.

It is true that there is a clear sense of continuity in my works. That said, the current manuscript starts where the pervious works end, and there is a zero overlap between the current work and published research. The major theoretical breakthrough of this work is the development of the asymptotic theory for an *arbitrary* number of colours, whereas the existing results are not fully covering the case of $N=2$.

3.

The result of color switching is interesting, but is also a property of very simple model the author proposed. The fact that his theory can be used to understand model well is nice, but the finding is property of the model. The text regarding this finding is also unclear.

The text discussing this phenomenon was improved. Additionally the Supplementary Section 12 now contains a parametric study suggesting that the switching may be a generic property of percolation in networks with communities. That said, this paper presents the colour switching as an observation, and no claim is made about generality here. Future analytical studies may shed more light on this remarkable feature of coloured networks. A similar explanation was added in the last paragraph on p. 13.

4.

Q1: Color switching is presented in simulations only for components of size 50. It is not really clear what happens in the giant component and what happens in finite components. I would assume that the whole fig 3 is for finite components,

as a number of links of type 3 is always greater than the number of links of type one (if we assume that the nodes of type 1 and type 3 have the same probability). So in Giant component for $p=1$ I would assume that the fraction of type 4 dominates, which is just the opposite to the figure (in which I would assume to have finite components dominated with type 1).

Size 50 has been used in Panel b of Fig. 5 to illustrate that the spread around the mean that occurs because of components being small can also be well predicted. The caption of Fig. 5 was modified to clarify this.

As Figure 3b shows, the mean colour fractions are the same in all finite components, and does not depend on component size. Yet, the giant component has a different colour structure. Eq.(16) was moved from SI to the main text to stress on this difference. Also explanatory text at the bottom of p. 13 was added to stress on this point.

The intuition behind what dominates is the following: we have a network with communities. The blue community is the most fragile, then goes red, and the yellow is most robust. (This follows from the choice of the degree distribution) The trend is thus as follows (right-to-left): the percolation chips off the fragile part first (blue dominates in small components), then second most fragile (red), and finally, the yellow community will dissociate.

5.

The presentation of critical manifolds is well written and understandable. This part of the manuscript is the one which would most influenced the field and the author should perhaps concentrate more effort on it. My problem with this part of manuscript is a certain “abuse” of language as author defines for small C_2 coefficients a “critical window” (notably for $C_2= 0.02$). While results are interesting I strongly urge author to change the language because what is a small value for C_2 is just arbitrary in this text. Unless there is a clear mathematical criterion introduced, author should avoid this language, especially given the fact that the whole text is generally very precise and stringent from mathematical point of view.

The terminology was changed ‘critical window’->‘critical interval’ to avoid the confusion. It is worth mentioning that the critical intervals are defined strictly for $C_2=0$. And one of the main findings is that such windows exist at all (i.e. they are wider than just critical points).

Small C_2 is discussed in reference to the fact that some networks might enter a region that is close to criticality (e.g. $C_2<0.02$) multiple times and therefore might *look* critical when found in small samples coming from empirical data. (This is the case in Ref. [24], for example). The text in the second paragraph of p. 13 has been updated to highlight this distinction.

6.

Q3: How it would relate to other percolation properties of complex networks. For example can this theory be related to the scaling laws as found in: "Dong, Gaogao, et al. "Resilience of networks with community structure behaves as if under an external field." Proceedings of the National Academy of Sciences (2018): 201801588."

This is a very relevant point and an additional analysis was performed to investigate it analytically. It was found that divergence of the expected component size is of $1/x$ type. See Supplementary Note 9, and also Eq. (20) in Results.

7.

Furthermore there were previous reports of critical manifolds (take for example: "Shao, Jia, et al. "Cascade of failures in coupled network systems with multiple support-dependence relations." Physical Review E 83.3 (2011): 036116." - can the author relate his theory to their findings?

There is indeed a large class of problems that employs the notion of strong connectivity. In contrast to this notion, the current manuscript utilises weak connectivity as the working definition for connected components. I personally think that strong connectivity (and thus, the cascades on interdependent networks) is a whole different world and I would not immediately see how to extend or relate the current approach to these problems. That said, there is a considerable progress in the field made with message passing algorithms and related ideas, and the reference provided by Reviewer is a good example of such approaches.

The beginning of Results (last paragraph on p.3) was extended to highlight the distinction between the weak and other types of connectivities

8.

There were also other approaches to model percolation of colored networks ("Krause, Sebastian M., et al."Color-avoiding percolation." Physical Review E 96.2 (2017): 022313.") does the approach presented here works in the case of this type of percolation?

When a few colours are avoided, this type of percolation relies upon strong notion of connectivity and therefore the same answer as in the previous comment applies. Whereas if one is interested in a single-colour avoidance, a link can be made to this work. The discussion of the connection to colour avoiding percolation was added to Results, Section B.

Additional Questions:

9.

Q2: There is a difference between EQ4 and Eq S7 in the supplement and I assume this are the same equations. Either D is an auxiliary function or is a matrix defined in S9 on which convolution determinant has to be computed.

The notation was updated.

10.

Q3: I do not really understand the equation 16, does it mean that $u(1,2,3)=\text{Poiss}(20,3)$? If so it would be nice to explain to the reader why you choose this particular unusual distribution. Is it just for ease of computations or to show that theory works also for correlated distributions?

This is just an example that yields a degree distribution with non-zero mixed moments (or alternatively with pair correlated degrees) The theory does not require the degree distribution to have a specific shape. This clarification was also added to the text below the equation.

11.

Q4: It would be nice if the author would provide a bit more of intuitive explanation to eq: 6 (it follows from analysis given in supplement, but one would like to get a feeling why the M matrix is so important for criticality). It is also possible that more intuitive explanation is not possible and in that case I would urge author to disregard this question.

It is hard to think of a physical meaning of the matrix M from the derivations present in the manuscript. Matrices of similar structure do occur in the message passing approach, and a reference to this fact was added to the last paragraph on p. 7.

Comments:

12.

C1: Author likes to use an uncommon notation which makes his text harder to follow. Excess degree distribution is biased degree distribution etc. This can in principle stay in the paper, but I would strongly suggest to the author to change his nomenclature to a more common to its expected audience (for example to use any of notations and nomenclature from any of the references 3-7).

The notation was updated

C2: Equation 6, 7 and 8 are very important and a feature of theory that SHOULD be stressed even more by the author.

The text highlighting the role of this equations in the theory was added on pp. 5-6. Also Fig.2 was added to introduce the concept of manifolds vs. critical points.

Reviewer #2:

1.

The manuscript seems to stand on a strong theoretical basis, but the application of this advance in the theory remains speculative: there are several statements where this theory could be used, but the examples presented in the manuscript do not illustrate examples of these speculative problems. If the theory can be used to solve outstanding problems in the literature or change the way some systems are analysed, why not show few examples of this? Further, comparison to previous theory is almost missing (except for a paragraph in the introduction and few reference later).

The text discussing connection to the prior studies were added (at the end of Sections A and B and also in Discussion (second paragraph on p.13)).

Additionally, Supplementary Note 10 was created specifically to discuss how other known problems can be viewed as essentially the edge-coloured problem. The author is of opinion, however, that the networks defined by their degree distributions have already become a standard object of study in networks science and its utility has been well covered in numerous prior empirical studies. For example in Refs [24, 25, 35, 37–40]. Just as we do not need to justify anymore why should one study spin glasses and take their utility for granted, the author believes that solely theoretical study of coloured networks justifies the merit. Of course, if Referee insists on adding more examples, the author will extend the manuscript accordingly.

2.

A. Connection to the previous literature could be stronger. First of all, it would be good to mention that these types of networks have been studied earlier under the name of multiplex networks. In fact, the concept "coloured networks" can be slightly misleading as it is often used also to denote networks where the nodes are coloured. Further, it would be good to mention the work on mutual percolations/cascading failures this related (but different) percolation process has been under active study in recent years.

There is indeed an important class of problems that employs the notion of strong connectivity. Introduction and the beginning of Results (last paragraph on p. 3) has been extended to highlight the distinction between the weak and strong connectivities and to

better root the current problem in the existing literature. Reference to multiplex networks were added accordingly in Introduction and the summarising paragraph of Discussion. Supplementary Note 10 was added to discuss that the node coloured problems is a subclass of edge coloured problems (although the converse is not true) which justifies the nomenclature.

3.

B. The connection of the theoretical results to previous very similar results are not explored properly. For example, Ref 16 defines a similar colored excess degree matrix to M and solves a very similar eigenvalue problem on it (see Eqs. 5 and 6 in ref 16, where the matrix B similar to M and an eigenvalue problem is defined). It is mentioned in subsection b that the results generalize the ones of equation 7 in ref 16, but the differences between the theories and their predictions should be made very clear to the reader.

This is indeed a very relevant distinction. A text highlighting this distinction was added to Results (last paragraph on p. 7), and also to Discussion, p. 13.

4.

C. Many real networks with multiple types of links contain large amounts of "edge overlap", where the same pairs of nodes are connected with multiple types of links. It seems that the formalism in this manuscript does not easily allow including this type of correlations in the model. This probably seriously limits the applicability of the results to many real-world systems, but could probably be introduced relatively easily to the equations(?). In any case this should be discussed in the manuscript.

It is indeed correctly mentioned that the overlap of edges is zero almost surely in the current framework. That said, one can recast the problem with edge overlap by using the idea of multilinks [Cellai et al. PhysRevE 2013] The discussion was append to Supplementary Note 10.

5.

D. The manuscript would be made stronger if the main results could be made more clear earlier. The sections A-F are rather dense and difficult to understand without following the references to the supplementary and methods, and things start to clear out only when the toy examples are shown in section F. Further, the results on the hierarchies of connectivities seem rather trivial as adding more links will always increase the connectivity, and the relevance and applicability of these results remain a mystery to me.

The manuscript has been restructured so that: 1. the anticipation of the main results is projected earlier in the text, 2. The justification behind why the hierarchies are important was made more explicit. Namely, that the secondary phase transition may bundle into clusters as governed by the Gershgorin circle theorem, and therefore there is a downward causation effect: the mutual criticality of a few layers together sets the boundary on when the criticality in each of this layers can occur. This explanation was also added to second paragraph on p. 8.

6.

Minor comments (mostly typos):

1. Page 6, "To this end,": *This is at the start of a paragraph, so it is not clear to what end.*

2. Page 6, "any colours form index set": *should it be "from"?*

3. Page 7, notation of eq 12?

4. Page 10, Fig 2 caption: "vales" -> "values"

5. Page 12, Fig 3 caption: "faction" -> "fraction"

Corrected.

REVIEWERS' COMMENTS:

Reviewer #1 (Remarks to the Author):

I am satisfied with changes done in the manuscript. It is more readable, the line of thought is easier to follow and I would recommend the publication of the manuscript in Nature Communications. Although significance of this paper by itself is borderline with respect to influence expected from papers published in Nature communications, I believe that the total body of which this work is direct continuation is significant and important enough (and still not recognized enough) that I have to recommend the publication of this paper. Personally, I will immediately use some of the ideas developed in this paper in my own research in percolation problems.

Reviewer #2 (Remarks to the Author):

In general the presentation of the article is now improved and the previous literature is now discussed to satisfactory extent. This article is a logical continuation of the authors theoretical work on connectivity of configuration model networks. It is not the first to consider connectivity in multiplex/edge-colored configuration networks, but thanks to the theoretical framework introduced in the earlier work of the author it is able to go further in some aspect than the previous work.

I also agree with the assessments made by the reviewer #1.

More detailed comments related to authors replies below.

1. Original comment:

The manuscript seems to stand on a strong theoretical basis, but the application of this advance in the theory remains speculative: there are several statements where this theory could be used, but the examples presented in the manuscript do not illustrate examples of these speculative problems. If the theory can be used to solve outstanding problems in the literature or change the way some systems are analysed, why not show few examples of this? Further, comparison to previous theory is almost missing (except for a paragraph in the introduction and few reference later).

Author Reply:

The text discussing connection to the prior studies were added (at the end of Sections A and B and also in Discussion (second paragraph on p.13).

Additionally, Supplementary Note 10 was created specifically to discuss how other known problems can be viewed as essentially the edge-coloured problem. The author is of opinion, however, that the networks defined by their degree distributions have already become a standard object of study in networks science and its utility has been well covered in numerous prior empirical studies. For example in Refs [24, 25, 35, 37–40]. Just as we do not need to justify anymore why should one study spin glasses and take their utility for granted, the author believes that solely theoretical study of coloured networks justifies the merit. Of course, if Referee insists on adding more examples, the author will extend the manuscript accordingly.

Reviewer reply:

The connection to the prior studies has now improved.

I agree that configuration models have been shown to be useful in the literature, and there is nothing wrong in them as a subject of a theoretical study. However, this is besides my point. The argument given by the author applies to any theoretical study on configuration models, no matter how minor or obscure. Using the analogy of the author, not everything written on the topic of spin classes should be published in journals such as Nature Communications. What matters is what are the improvements of this study to the state of the art of the field, not that the study is on particular field.

My original comment was a suggestion how to make the manuscript stronger by going from purely mathematical paper to one clearly illustrating how the new theory can be used to increase the understanding on real systems in a way that was not possible with previous theoretical tools. I am not insisting on any additions, but my original assesment still holds. This is an article on mathematics of percolation with only very vague speculations on the applications of the theory as compared to previous methods. This is not to say that I would think that this theory, especially when considering the whole line of work of the author in his previous papers, cannot become useful or even a standard tool in network science.

2. Original comment

Connection to the previous literature could be stronger. First of all, it would be good to mention that these types of networks have been studied earlier under the name of multiplex networks. In fact, the concept "coloured networks" can be slightly misleading as it is often used also to denote networks where the nodes are coloured. Further, it would be good to mention the work on mutual percolations/ cascading failures this related (but different) percolation process has been under active study in recent years.

Author reply:

There is indeed an important class of problems that employs the notion of strong connectivity. Introduction and the beginning of Results (last paragraph on p. 3) has been extended to highlight the distinction between the weak and strong connectivities and to

better root the current problem in the existing literature. Reference to multiplex networks were added accordingly in Introduction and the summarising paragraph of Discussion. Supplementary Note 10 was added to discuss that the node coloured problems is a subclass of edge coloured problems (although the converse is not true) which justifies the nomenclature.

Reviewer reply:

The manuscript has improved in making connection with the previous literature. "Strong connectivity" (also known with many other names) is now mentioned shortly.

The nomenclature becomes clear once you read the article and the note 10 from the supplementary information section. For the title it would be nice to have as little ambiguity as possible. This is of course very difficult, because different research communities use different language. This is also why I don't want to make any strict recommendations on the nomenclature, but just to point out the this should be carefully thought through. It would make the article more approachable by larger readership if the connection to the various naming conventions would be at least very shortly mentioned in the introduction. Further, if the note 10 was referred already in the introduction this might also be helpful for the reader who is working with these specific types of networks.

Here are few minor notes related to the presentation:

- Page 2, third paragraph: "configuration multiplex": What exactly is meant by this. It would be a good to have at least a citation where this term is used.

- Page 3, "stronger predictive power": Is this related to the concept of predictive power in statistics? In this context it is not clear what is meant by this.

- Page 3, "In most of these studies, however, the theoretical side is often less developed as it has to be scattered between special cases to explain empirical observations rather than to develop a single universal theory." : It would be helpful to say which ones are the ones that develop a single universal theory and which are not instead of citing evrything together.

- Page 4, "Note, there are many other ways to define a connecting path": Apart from the last citation the other papers use the same type of connectivity (formalized in a different ways, but they could still be mapped into the same notion of connectivity). This has caused a lot of confusion in the literature earlier on, and this sentence only adds to this confusion.

- Page 7, "configurational multi-graphs" : do you mean edge-colored multi-graphs? As there are several terminologies used in the field it could be good to refer this with multiple names to cater to as wide audience as possible (e.g., multiplex networks with edge overlap)

3. Original comment

The connection of the theoretical results to previous very similar results are not explored properly. For example, Ref 16 defines a similar colored excess degree matrix to M and solves a very similar eigenvalue problem on it (see Eqs. 5 and 6 in ref 16, where the matrix B similar to M and an eigenvalue problem is defined). It is mentioned in subsection b that the results generalize the ones of equation 7 in ref 16, but the differences between the theories and their predictions should be made very clear to the reader.

Author reply:

This is indeed a very relevant distinction. A text highlighting this distinction was added to Results (last paragraph on p. 7), and also to Discussion, p. 13.

Reviewer reply:

This aspect was improved a lot.

Maybe a slight clarification would be useful for the reader: The Eq. 12 derived by Ref. [44] and Ref. [24] can be used when M is primitive matrix, which as is said later in the manuscript corresponds to the case when the network is disjoint. That is, in this case one could analyse the disjoint parts separately using Eq. 12?

4. Original comment

Many real networks with multiple types of links contain large amounts of "edge overlap", where the same pairs of nodes are connected with multiple types of links. It seems that the formalism in this manuscript does not easily allow including this type of correlations in the model. This probably seriously limits the applicability of the results to many real-world systems, but could probably be introduced relatively easily to the equations(?). In any case this should be discussed in the manuscript.

Author reply:

It is indeed correctly mentioned that the overlap of edges is zero almost surely in the current framework. That said, one can recast the problem with edge overlap by using the idea of multilinks [Cellai et al. PhysRevE 2013] The discussion was added to Supplementary Note 10.

Reviewer reply:

This issue is now addressed in a very minimal way. Of course one needs to also adjust the formulas related to percolation where a fraction of edges are removed (in a way that you don't remove double edges), hierarchies, colors of edges in giant component etc., which is probably going to be

relatively easy but technical task. Further, this approach has the same limitations as discussed in the Cellai et al. article: the M matrix will very quickly blow up when the number of edge types grows, which can make the analysis difficult in practice.

The numbering related to the references to the supplementary notes seems to be off by one.

5. Original comment

The manuscript would be made stronger if the main results could be made more clear earlier. The sections A-F are rather dense and difficult to understand without following the references to the supplementary and methods, and things start to clear out only when the toy examples are shown in section F. Further, the results on the hierarchies of connectivities seem rather trivial as adding more links will always increase the connectivity, and the relevance and applicability of these results remain a mystery to me.

Author reply:

The manuscript has been restructured so that: 1. the anticipation of the main results is projected earlier in the text, 2. The justification behind why the hierarchies are important was made more explicit. Namely, that the secondary phase transition may bundle into clusters as governed by the Gershgorin circle theorem, and therefore there is a downward causation effect: the mutual criticality of a few layers together sets the boundary on when the criticality in each of this layers can occur. This explanation was also added to second paragraph on p. 8.

Reviewer reply:

The overall presentation of the manuscript has now improved. The structure of the manuscript is rather unusual with the detailed analysis of the toy examples appearing in the discussion section.

The discussion on the hierarchies has also improved slightly. The focus is now more in the Gershgorin circle theorem, and there are now few sentences on the interpretation of this result. This didn't clear up the reason why this is brought up: is this a solution for a known problem or a solution looking for a problem? Based on the current presentation of the result, it seems to that the latter description is more accurate.

There is now also an attempt to connect to the previous literature on color-avoiding percolation, although (as acknowledged in the manuscript) this is only a similar problem, not exactly the same problem.

REVIEWERS' COMMENTS:

Reviewer #1 (Remarks to the Author):

I am satisfied with changes done in the manuscript. It is more readable, the line of thought is easier to follow and I would recommend the publication of the manuscript in Nature Communications. Although significance of this paper by itself is borderline with respect to influence expected from papers published in Nature communications, I believe that the total body of which this work is direct continuation is significant and important enough (and still not recognized enough) that I have to recommend the publication of this paper. Personally, I will immediately use some of the ideas developed in this paper in my own research in percolation problems.

Author Reply:

The author would like to thank the referee for numerous valuable suggestions, which to a great extent helped to improve the manuscript.

Reviewer #2 (Remarks to the Author):

In general the presentation of the article is now improved and the previous literature is now discussed to satisfactory extent. This article is a logical continuation of the authors theoretical work on connectivity of configuration model networks. It is not the first to consider connectivity in multiplex/edge-colored configuration networks, but thanks to the theoretical framework introduced in the earlier work of the author it is able to go further in some aspect than the previous work.

I also agree with the assessments made by the reviewer #1.

More detailed comments related to authors replies below.

1. Original comment:

The manuscript seems to stand on a strong theoretical basis, but the application of this advance in the theory remains speculative: there are several statements where this theory could be used, but the examples presented in the manuscript do not illustrate examples of these speculative problems. If the theory can be used to solve outstanding problems in the literature or change the way some systems are analysed, why not show few examples of this? Further, comparison to previous theory is almost missing (except for a paragraph in the introduction and few reference later).

Author Reply:

The text discussing connection to the prior studies were added (at the end of Sections A and B and also in Discussion (second paragraph on p.13)).

Additionally, Supplementary Note 10 was created specifically to discuss how other known problems can be viewed as essentially the edge-coloured problem. The author is of opinion, however, that the networks defined by their degree distributions have already become a standard object of study in networks science and its utility has been well covered in numerous prior empirical studies. For example in Refs [24, 25, 35, 37–40]. Just as we do not need to justify anymore why should one study spin glasses and take their utility for granted, the author believes that solely theoretical study of coloured networks justifies the merit. Of course, if Referee insists on adding more examples, the author will extend the manuscript accordingly.

Reviewer reply:

The connection to the prior studies has now improved.

I agree that configuration models have been shown to be useful in the literature, and there is nothing wrong in them as a subject of a theoretical study. However, this is besides my point. The argument given by the author applies to any theoretical study on configuration models, no matter how minor or obscure. Using the analogy of the author, not everything

written on the topic of spin classes should be published in journals such as Nature Communications. What matters is what are the improvements of this study to the state of the art of the field, not that the study is on particular field.

My original comment was a suggestion how to make the manuscript stronger by going from purely mathematical paper to one clearly illustrating how the new theory can be used to increase the understanding on real systems in a way that was not possible with previous theoretical tools. I am not insisting on any additions, but my original assesment still holds. This is an article on mathematics of percolation with only very vague speculations on the applications of the theory as compared to previous methods. This is not to say that I would think that this theory, especially when considering the whole line of work of the author in his previous papers, cannot become useful or even a standard tool in network science.

Author Reply:

I share the opinion that, when such a possibility exists, it is necessary to demonstrate the a new theory on real world problems. The theory presented in this manuscript is exactly the theory of a great applied potential. That said, in view of length limitations and limitations of complexity that a single publication has to address without compromising on clarity, I would opt to postpone the opportunity of such a demonstration to my forthcoming publications.

2. Original comment

Connection to the previous literature could be stronger. First of all, it would be good to mention that these types of networks have been studied earlier under the name of multiplex networks. In fact, the concept "coloured networks" can be slightly misleading as it is often used also to denote networks where the nodes are coloured. Further, it would be good to mention the work on mutual percolations/ cascading failures this related (but different) percolation process has been under active study in recent years.

Author reply:

There is indeed an important class of problems that employs the notion of strong connectivity. Introduction and the beginning of Results (last paragraph on p. 3) has been extended to highlight the distinction between the weak and strong connectivities and to better root the current problem in the existing literature. Reference to multiplex networks were added accordingly in Introduction and the summarising paragraph of Discussion. Supplementary Note 10 was added to discuss that the node coloured problems is a subclass of edge coloured problems (although the converse is not true) which justifies the nomenclature.

Reviewer reply:

The manuscript has improved in making connection with the previous literature. "Strong connectivity" (also known with many other names) is now mentioned shortly.

The nomenclature becomes clear once you read the article and the note 10 from the supplementary information section. For the title it would be nice to have as little ambiguity as possible. This is of course very difficult, because different research communities use different language. This is also why I don't want to make any strict recommendations on the nomenclature, but just to point out the this should be carefully thought through. It would make the article more approachable by larger readership if the connection to the various naming conventions would be at least very shortly mentioned in the introduction. Further, if the note 10 was referred already in the introduction this might also be helpful for the reader who is working with these specific types of networks.

Here are few minor notes related to the presentation:

1. Page 2, third paragraph: "configuration multiplex": What exactly is meant by this. It would be a good to have at least a citation where this term is used.

2. Page 3, "stronger predictive power": Is this related to the concept of predictive power in statistics? In this context it is not clear what is meant by this.

3. Page 3, "In most of these studies, however, the theoretical side is often less developed as it has to be scattered between special cases to explain empirical observations rather than to develop a single universal theory." : It would be helpful to say which ones are the ones that develop a single universal theory and which are not instead of citing evreything together.

4. Page 4, "Note, there are many other ways to define a connecting path": Apart from the last citation the other papers use the same type of connectivity (formalized in a different ways, but they could still be mapped into the same notion of connectivity). This has caused a lot of confusion in the literature earlier on, and this sentence only adds to this confusion.

5. Page 7, "configurational multi-graphs" : do you mean edge-colored multi-graphs? As there are several terminologies used in the field it could be good to refer this with multiple names to cater to as wide audience as possible (e.g., multiplex networks with edge overlap)

Author reply:

Word multiplex was added to the title of the manuscript, also a reference to Supplementary Note 1 (former Note 10) was made in the introductory text of the edge-coloured model.

Reply to minor notes:

1. References, as well as an alternative nomenclature was added.
2. wording was adjusted to instead use "better predictions".
3. corrected.
4. The text was adjusted: "there are several other ways to define a connected component...'", also the references are cited separately now.
5. The text was adjusted as advised.

3. Original comment

The connection of the theoretical results to previous very similar results are not explored properly. For example, Ref 16 defines a similar colored excess degree matrix to M and solves a very similar eigenvalue problem on it (see Eqs. 5 and 6 in ref 16, where the matrix B similar to M and an eigenvalue problem is defined). It is mentioned in subsection b that the results generalize the ones of equation 7 in ref 16, but the differences between the theories and their predictions should be made very clear to the reader.

Author reply:

This is indeed a very relevant distinction. A text highlighting this distinction was added to Results (last paragraph on p. 7), and also to Discussion, p. 13.

Reviewer reply:

This is aspect was improved a lot.

Maybe a slight clarification would be useful for the reader: The Eq. 12 derived by Ref. [44] and Ref. [24] can be used when M is primitive matrix, which as is said later in the manuscript corresponds to the case when the network is disjoint. That is, in this case one could analyse the disjoint parts separately using Eq. 12?

Author Reply:

I agree with the remark. Although, the most peculiar situation is when an originally primitive matrix is perturbed in such a way that it fails to be primitive. The clarification was added to the manuscript as suggested.

4. Original comment

Many real networks with multiple types of links contain large amounts of "edge overlap", where the same pairs of nodes are connected with multiple types of links. It seems that the formalism in this manuscript does not easily allow including this type of correlations in the

model. This probably seriously limits the applicability of the results to many real-world systems, but could probably be introduced relatively easily to the equations(?). In any case this should be discussed in the manuscript.

Author reply:

It is indeed correctly mentioned that the overlap of edges is zero almost surely in the current framework. That said, one can recast the problem with edge overlap by using the idea of multilinks [Cellai et al. PhysRevE 2013] The discussion was append to Supplementary Note 10.

Reviewer reply:

This issue is now addressed in a very minimal way. Of course one needs to also adjust the formulas related to percolation where a fraction of edges are removed (in a way that you don't remove double edges), hierarchies, colors of edges in giant component etc., which is probably going to be relatively easy but technical task. Further, this approach has the same limitations as discussed in the Cellai et al. article: the M matrix will very quickly blow up when the number of edge types grows, which can make the analysis difficult in practice.

The numbering related to the references to the supplementary notes seems to be off by one.

Author Reply:

In order to apply the current theory to a new type of a network, as for instance networks with edge multiplicity and overlaps, one needs to formulate the corresponding master equation and extract the partial moments of the effective coloured degree distribution,

which provide the input for the current theory. The overlap or multiplicity thus do not pose a conceptually new paradigm. I agree with the referee that such a task is relatively easy but it is also a technical one. The Supplementary Note 1 was extended to include a short discussion on how such master equation can be composed for the edge multiplicity case. That said, a complete coverage of this topic together with an analysis goes beyond the format of the supplementary note and certainly requires a separate publication.

The numbering of supplementary notes was reviewed.

5. *Original comment*

The manuscript would be made stronger if the main results could be made more clear earlier. The sections A-F are rather dense and difficult to understand without following the references to the supplementary and methods, and things start to clear out only when the toy examples are shown in section F. Further, the results on the hierarchies of connectivities seem rather trivial as adding more links will always increase the connectivity, and the relevance and applicability of these results remain a mystery to me.

Author reply:

The manuscript has been restructured so that: 1. the anticipation of the main results is projected earlier in the text, 2. The justification behind why the hierarchies are important was made more explicit. Namely, that the secondary phase transition may bundle into clusters as governed by the Gershgorin circle theorem, and therefore there is a downward causation effect: the mutual criticality of a few layers together sets the boundary on when the criticality in each of this layers can occur. This explanation was also added to second paragraph on p. 8.

Reviewer reply:

The overall presentation of the manuscript has now improved. The structure of the manuscript is rather unusual with the detailed analysis of the toy examples appearing in the discussion section.

The discussion on the hierarchies has also improved slightly. The focus is now more in the Gershgorin circle theorem, and there are now few sentences on the interpretation of this result. This didn't clear up the reason why this is brought up: is this a solution for a known problem or a solution looking for a problem? Based on the current presentation of the result, it seems to that the latter description is more accurate.

There is now also an attempt to connect to the previous literature on color-avoiding percolation, although (as acknowledged in the manuscript) this is only a similar problem, not exactly the same problem.

Author Reply:

Concerning the distribution of secondary critical points as derived from Gershgorin circle theorem, I think this finding provides sufficient scientific merit as a theoretical results in itself. This result gives a precise quantification behind a very intuitive statement: percolation points of separate layers define the percolation point of the overall system. Moreover, the findings also state that this intuitive statement does not hold if matrix M lacks diagonal-dominance, i.e. when the number mixed-degree nodes is small.

The author would like to thank the referee for numerous valuable suggestions, which to a great extent helped to improve the manuscript.